# Climate change impacts on ocean light in Arctic ecosystems

Trond Kristiansen [1,2] ✉, Øystein Varpe [3,4], Elizabeth R. Selig [5], Benjamin J. Laurel [6], William J. Sydeman [1], Michaela I. Hegglin [7,8] & Phillip J. Wallhead [9]

Climate change is causing major sea ice losses, leading to increased light availability across polar marine ecosystems, however the consequences are largely unknown. We quantify how future conditions for sea ice and snow, storm-driven waves, clouds, ozone, air and ocean temperature, and chlorophyll-a will affect seasonal absorption and reflection of light in Arctic seas, alongside growth and survival of fish. Using four CMIP6 model inputs and a spectral radiative transfer model, we predict a 75–160% increase in visible light by 2100 in the Northern Bering, Chukchi, and Barents Seas. We predict increased sunlight and warmer summer waters, with reduced phytoplankton levels, will negatively impact cold-water fish species growth and survival during summer, demonstrated here for polar cod. Asynchrony in prey and light availability, with prolonged periods of warmer waters, will reduce polar cod survival in the fall and restrict habitats in these regions after 2060. Warmer-water species like walleye pollock and Atlantic cod will be less impacted but may struggle at high latitudes during the polar night. Ocean warming coupled with increased light availability will accelerate changes in Arctic ecosystems, compromising the growth and survival of Arctic species in transitional zones and facilitating the northward expansion of boreal species.

Light is the primary driver for life in the world's oceans but has been under-appreciated in assessments of climate change impacts on marine ecosystem[1–6]. Changes in the optical properties of the ocean's skin regulate light penetration into the water column and influence a range of ecological processes such as the seasonal timing and amount of biological production[7], species distribution[8], and light-driven behavior and feeding[9]. Climate change-driven changes in the distribution, concentration, and thickness of sea ice, including the possibility of ice-free summers in the Arctic by 2050[10], portend a dramatic shift in seasonal light availability in polar ecosystems[11]. Given these predicted changes and the structuring role of light on Arctic species, a better understanding of changes in the quantity of seasonal light and subsequent impacts on ecological interactions is required to predict how future polar ecosystems may function.

We quantify how climate change will affect light regimes in the Arctic using a new approach that allows for the detailed analysis of the large-scale spectral changes in shortwave radiation under climate change. To this end we combine individually published radiative transfer model (RTM) algorithms that are forced by CMIP6 climate model outputs and used to quantify the spectral albedo from waves and chlorophyll[12,13], albedo from snow and ice[12], and the spectral attenuation of light moving through clouds[14], ozone[15], ice[16,17], snow[18],

[1]Farallon Institute, Petaluma, USA. [2]Actea Inc, San Francisco, California, USA. [3]Department of Biological Sciences, University of Bergen, and Bjerknes Centre for Climate Research, Bergen, Norway. [4]Norwegian Institute for Nature Research, Bergen, Norway. [5]Stanford Center for Ocean Solutions, Stanford University, Stanford, USA. [6]Alaska Fisheries Science Center, NOAA, Newport, USA. [7]Institute of Climate and Energy Systems – Stratosphere (ICE-4), Forschungszentrum Jülich, Julich, Germany. [8]Department of Meteorology, University of Reading, Reading, UK. [9]Norwegian Institute for Water Research, Oslo, Norway. ✉ e-mail: trondkr@faralloninstitute.org

and chlorophyll-a[19] (Fig. 1). We selected two climate futures for the forcing data, the Shared Socioeconomic Pathways (SSP) SSP2-4.5 and SSP5-8.5[20]. Together, they represent the intermediate and extreme ranges of climate futures (figures show generally the SSP2-4.5 results, while SSP5-8.5 results, unless otherwise indicated, can be found in the supplementary). This study used outputs from four independent CMIP6 models across a range of model realizations, a total of 16 combinations (Supplementary Table 2), to quantify light. We combine RTM outputs using forcing from models with varying strengths[21,22] to create an ensemble that is representative for model variability: the MPI-ESM1-2-LR model captures the trends in Arctic sea ice extent and thickness[22], MPI-ESM1-2-HR has demonstrated strong ability to simulate Arctic conditions sea ice conditions[21,23,24], UKESM1-0-LL has shown strong skills in projecting sea ice dynamics[23,25], and CanESM5 has notable skill in decadal climate predictions[26] and good ability to simulate Arctic Sea ice variability[23,25]. The RTM calculates direct normal irradiance (DNI) and diffuse horizontal irradiance (DHI) and we estimate the spectral albedo i) over the open ocean as a function of ocean waves and chlorophyll-a concentration, latitude, time of day and year, and ii) over sea ice and snow-covered seas. After accounting for reflection, the model attenuates the remaining light spectrally as it moves through the snow layer, sea ice, and chlorophyll-a. A sensitivity test of the RTM provided insights into how the individual components

and parameterizations of the model impacted the outputs, and we have confidence in the model's results based on these findings. The RTM results allow us to estimate the spectrally integrated light that reaches the upper part of the water column within the Arctic Ocean.

Here, we focus on how light availability is expected to change within the Northern Bering and Chukchi and the Barents Seas[27], which represent two highly biologically productive, but rapidly changing, seasonally sea ice covered ecosystems. With these results, we analyze changes in "visible light" or photosynthetic active radiation (PAR; 400–700 nm), which is an essential component of photosynthesis[28]. The amount of light reaching the water column also elevates ocean temperatures. We then analyze the spatially integrated temporal variability of light and temperature across geographic domains. We also examine UV-B light (280–315 nm), which can be a significant stressor for phytoplankton[29], zooplankton[30], and fish embryos[31], and affects the development and growth of larval fish[32]. By comparing how these variables will change across the Northern Bering and Chukchi Sea and the Barents Sea large marine ecosystems (LMEs), we can estimate the consequences of shifting light and temperature regimes on the egg and juvenile stages of three abundant pelagic fish species in Arctic and sub-Arctic seas: polar cod (*Boreogadus saida*), walleye pollock (*Gadus chalcogrammus*), and Atlantic cod (*Gadus morhua*).

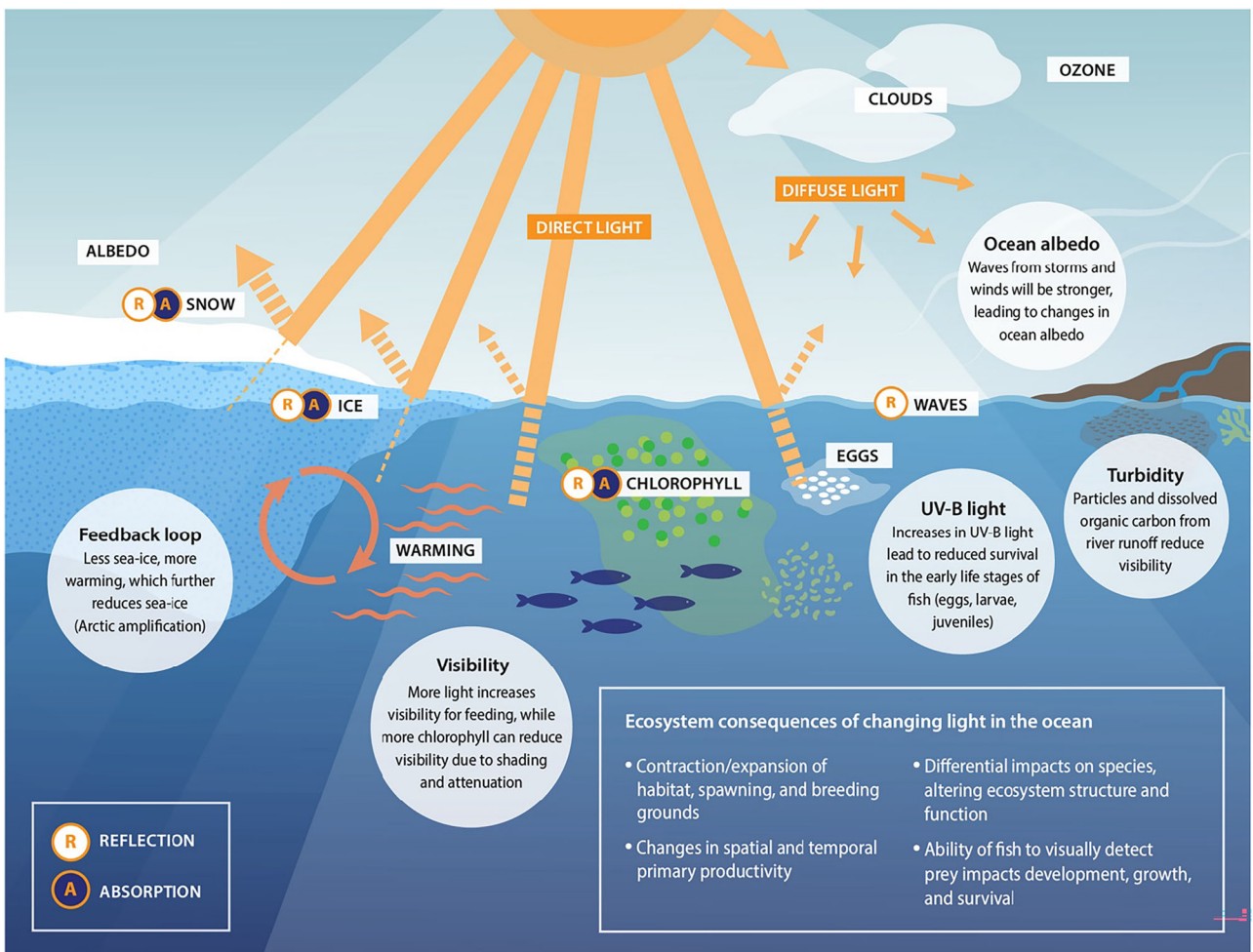

**Fig. 1 | Climate change impacts on the ocean light.** Future light regimes were modelled based on physical changes in ozone, clouds, waves, ice, snow and melt ponds as well as biological changes in chlorophyll, which all will impact reflection and absorption. We account for both direct and diffuse light from incoming shortwave radiation, albedo/reflection, and absorption. Ecosystem consequences of changing light are driven by a range of underlying mechanisms (circles). Most fish are visual feeders and require light to find their prey, while the development of fish eggs is temperature dependent. Phytoplankton particles can attenuate light, reducing visibility at depth (See Methods for details on the radiative transfer model).

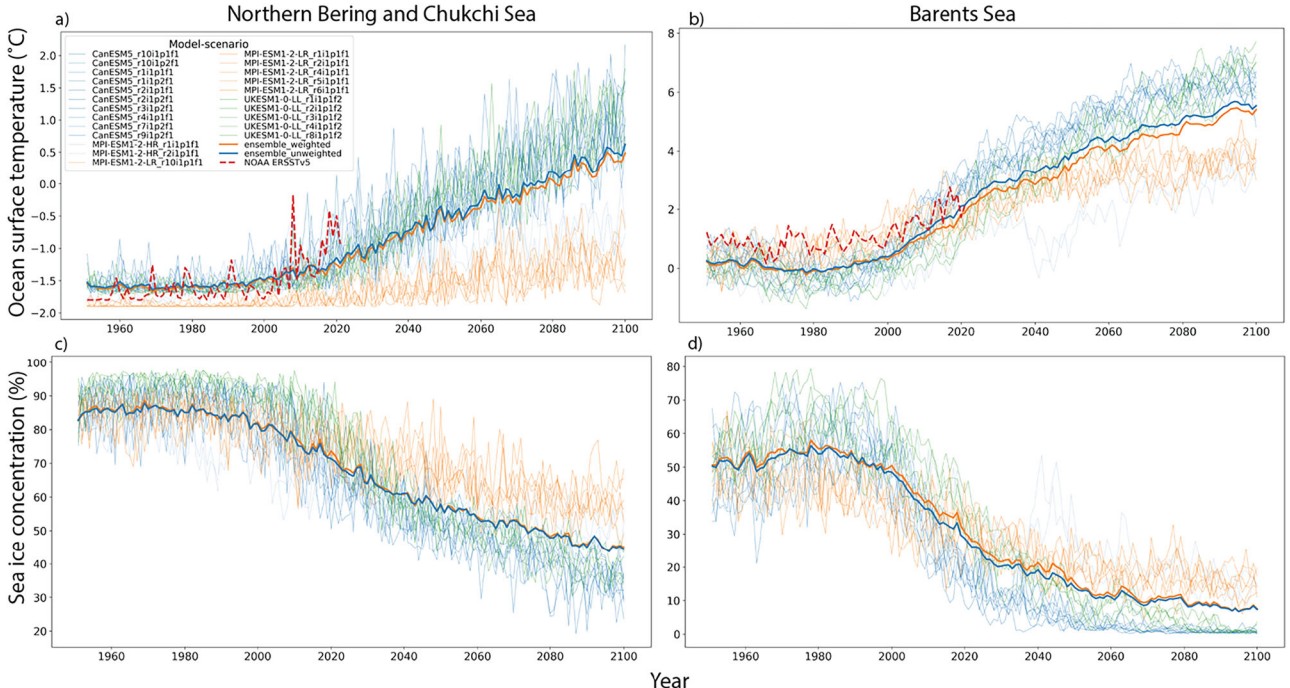

**Fig. 2 | Projected changes in ocean surface temperature and sea ice concentration.** Expected changes in ocean surface temperature (°C) and sea ice concentration (%) for (**a, c**) the Northern Bering and Chukchi Sea and (**b, d**) the Barents Sea large marine ecosystems for climate change scenario SSP2-4.5. Projections are shown for each of the individual CMIP6 model and realization combinations used as forcing for the RTM. Thick blue and orange lines show the ensemble average without and with weighting based on skill and independence. Red dashed line (**a, b**) shows observed sea surface temperature from NOAA ERSSTv5 dataset[67]. Projected changes under SSP5-8.5 are shown in Supplementary Fig. 3.

Polar cod (or Arctic cod in the U.S.) is a bellwether species for assessing the impacts of changing light regimes on Arctic ecosystems because of their roles in trophic transfer, circumpolar distribution, and life history traits that are adapted to polar conditions[33,34]. Polar cod also spawn their eggs under or near sea ice, where they are protected from breaking waves and UV-B radiation in the surface layer[32]. Walleye pollock and Atlantic cod have adjacent or overlapping populations and are also abundant and commercially important[34,35], but are more adapted to temperate conditions and therefore may be affected differently by changing light conditions. For all three of these fish species, the degree to which the timing of egg hatching coincides with phytoplankton and zooplankton blooms affects their immediate survival[36], population dynamics[37–39], and the potential energy transfer from zooplankton to higher trophic levels, such as seabirds and mammals[33]. For this study, we use climate model projections of chlorophyll-a as a proxy for food of planktivorous fish such as Polar cod and the larvae and juvenile stages of walleye pollock and Atlantic cod. Light calculations using the RTM provided us with ensemble estimates of seasonal variability of the euphotic conditions in the marine environment for two climate scenarios across 16 individual CMIP6 models and realizations. Our analyses highlight how warming and changing light availability impact the feeding, growth, and survival potential of these species, and with this information we illustrate potential consequences across Arctic food webs.

## Results

### Changes in photosynthetic active radiation (PAR) and its drivers

Overall, the projected decreases in sea ice and snow extent and thickness (Fig. 2, Supplementary Figs.1–3) will lead to elevated light conditions (Fig. 3, Supplementary Fig. 4). Enhanced light availability in the water column will then lead to increased ocean water temperatures (Fig. 2) and an extended open water season. The annual average area with open water within the Northern Bering and Chukchi and Barents Seas will increase from 50–55% in 1980 to 70–95% by 2100, depending on the climate scenario. We also estimate that the total area where light levels will exceed the minimum threshold[40] for fish feeding of 0.1 W/m$^2$ (Laurel pers. comm.) will increase by 25–30% and 14–16% in the Northern Bering and Chukchi and Barents Seas, respectively. Our choice of light threshold is based on one study and we realize that more research is needed to fully understand the effect of light levels on feeding. Based on climate scenarios SSP2-4.5 and SSP5-8.5 (Fig. 3, Supplementary Fig. 3), visible light (PAR) reaching the surface water column will increase by 55–160% annually by the year 2100 in response to the increased fraction of open water, mainly driven by reduced sea ice concentration (Fig. 2, Supplementary Figs. 1–3), as well as other changes in physical factors simulated in the CMIP6 future model runs (e.g., changes in cloud cover, increased air temperatures, reduced sea ice thickness). Reductions in snow and sea ice thickness and increased melt pond area due to warmer air temperatures will also contribute to more light entering the water column (Supplementary Figs. 1, 5).

Together these physical conditions will further increase the heat content of surface waters through the ice-ocean albedo feedback loop[41], leading to greater ice–melting, a physical dynamic unrelated to cloud cover. For both the Northern Bering and Chukchi Sea and Barents Sea, the average cloud cover remains nearly constant over the time period 1979–2100, at around 82% and 87%, respectively (Supplementary Fig. 2). An increase of 2% is found for the SSP5-8.5 scenario in the Northern Bering and Chukchi Sea. These results suggest that changes in cloud cover are not a main driver of changes in PAR in the ocean, compared to the reduced extent and thickness of snow and sea ice. By 2050, annual average PAR is estimated to increase by 50–77% within each region compared to 1980–2000, with continued increases of 0.009–0.016 W m$^{-2}$ y$^{-1}$ (Fig. 3) until 2100. Seasonal strong increases in PAR, particularly between April and September, are expected under both SSP2-4.5 and SSP5-8.5 (Fig. 3), while the winter months will continue to be relatively dark with only minor changes (Fig. 3). We also

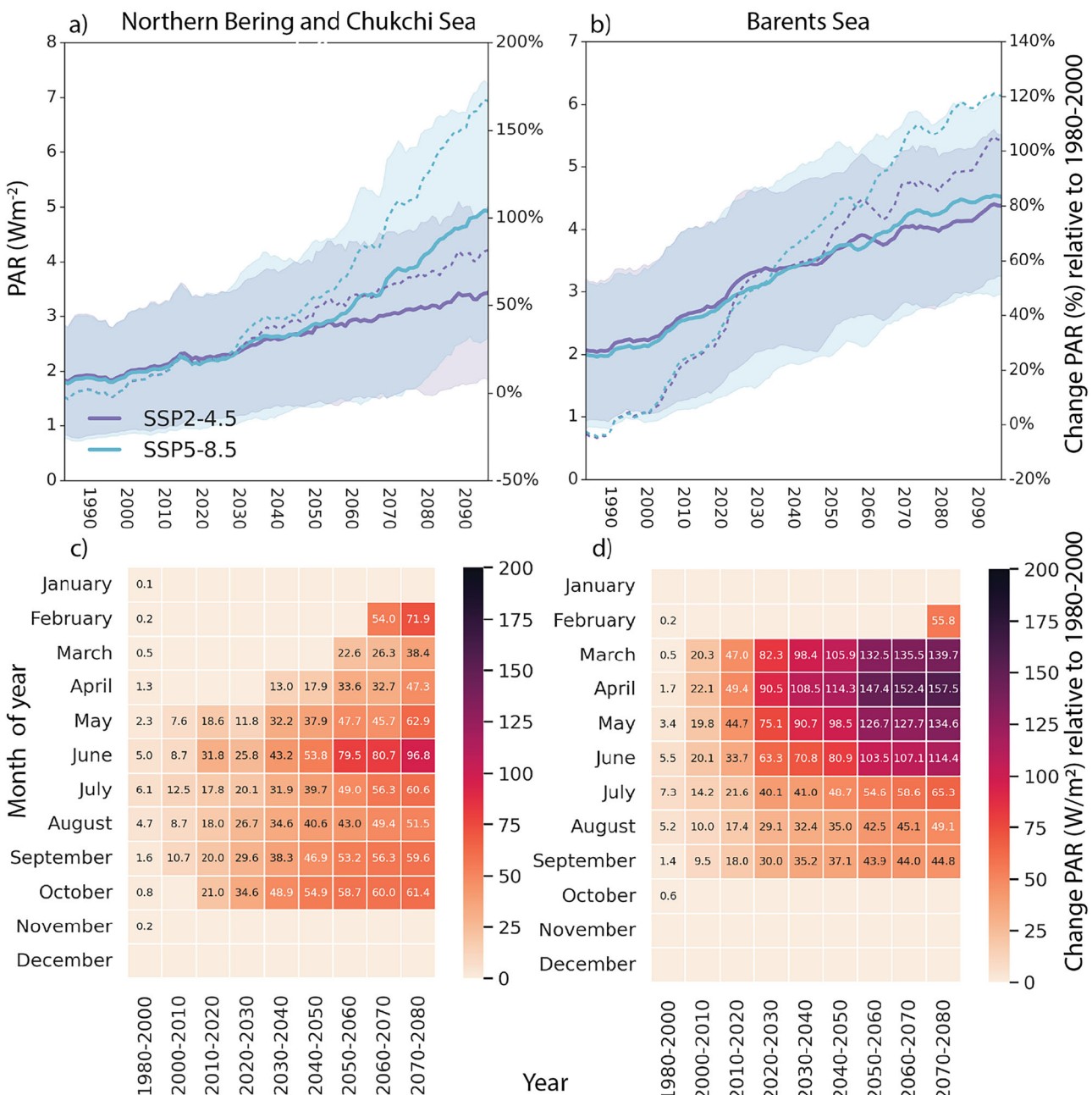

**Fig. 3 | Expected temporal changes in Photosynthetically Active Radiation (PAR).** Changes in visible (PAR, Wm⁻²) annual average light in the ocean for (**a**) the Northern Bering and Chukchi Sea and (**b**) the Barents Sea large marine ecosystems for two scenarios of climate change, SSP2-4.5 and SSP5-8.5. Average values are shown as solid lines with 95% confidence intervals in shading. Values are averaged across the LMEs with a 5-year rolling mean. The dashed lines show the relative change in % of PAR under SSP2-4.5 and SSP5-8.5 relative to the 1980–2000 average, with the scale on the right. Heatmaps showing the historical (1980–2000) monthly mean (leftmost column of each heatmap) and future decadal average changes in PAR (Wm⁻²) for (**c**) Northern Bering and Chukchi and (**d**) Barents Seas. Changes relative to historical values (1980–2000) are shown for each decade between 2000 and 2090 at 10-year intervals. Months with no change relative to historical values have no annotated values. Heatmaps depict results for SSP2-4.5, while Supplementary Fig. 4 shows SSP5-8.5.

observe a sudden significant increase in PAR after 2050 for the SSP5-8.5 scenario in the Northern Bering and Chukchi Sea. This could indicate that the system will then reach a tipping point potentially caused by the collapse of sea ice concentration and a strong increase in air and ocean temperatures.

## UV-B under climate change
Overall, we find only small changes in the average UV-B irradiance reaching the atmosphere-ice/ocean interface relative to 1980–2000 for both ecosystems (Fig. 4, top panels), although with pronounced

decadal variability. UV-B levels from 1980 to 2100 are generally decreasing, with a range of 0 to −10% for SSP5-8.5 and −2% to +8% for SSP2-4.5. The UV-B decline also has a strong seasonal component (not shown), with the largest and most distinct changes in total column ozone (TCO) during late winter/spring and the smallest changes during summer, while autumn has large variability. The decrease in annually averaged UV-B irradiance is driven by increased TCO, consistent with the future recovery of the stratospheric ozone layer following the Montreal Protocol regulations[15]. The stronger increase is found under the SSP5-8.5 scenario (Fig. 4, bottom panels), with TCO

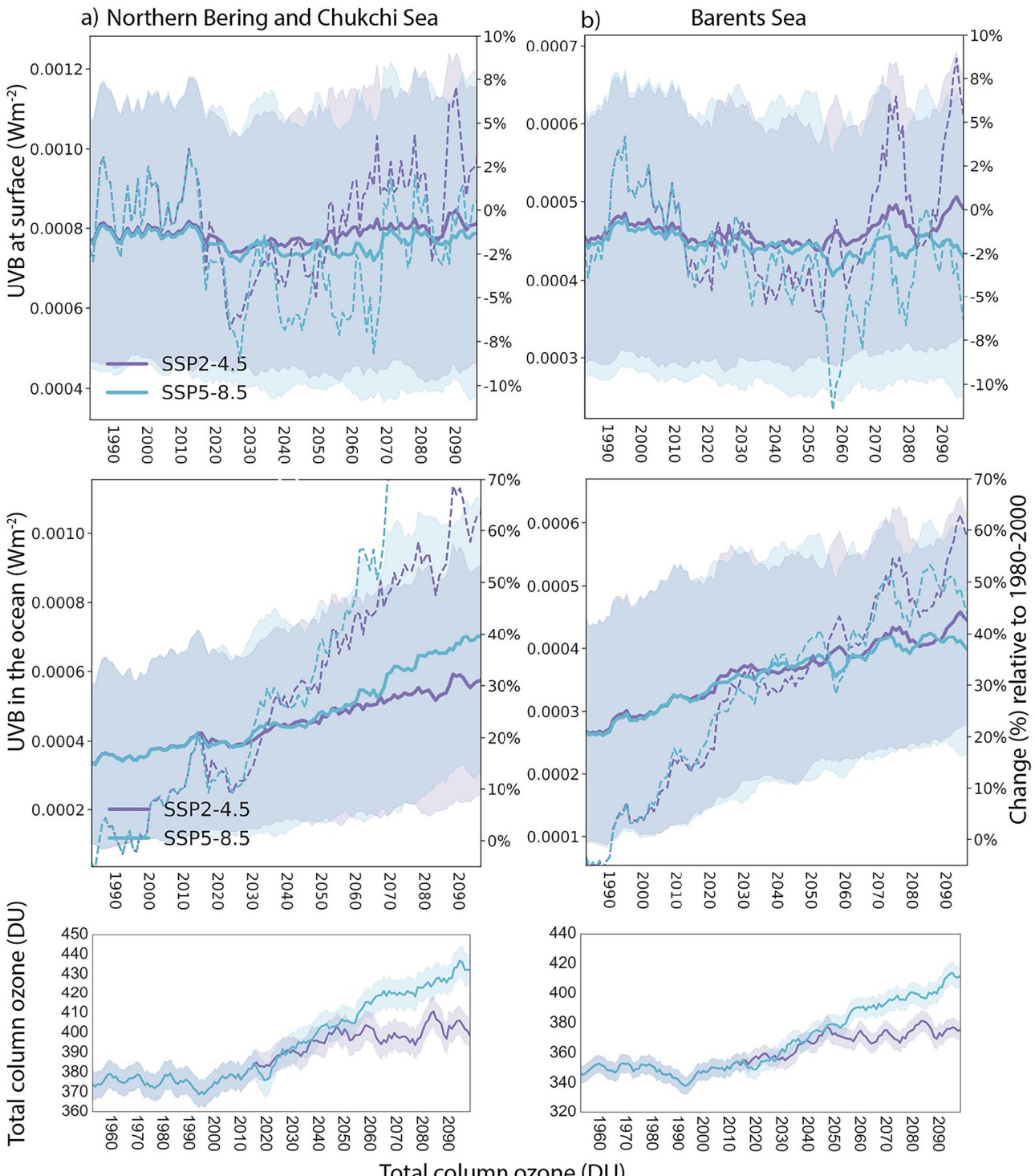

**Fig. 4 | Expected temporal changes in UV-B and TCO.** Projected changes in incoming UV-B (Wm⁻²) radiation at the atmosphere-ice/ocean interface are shown in top panel, while irradiance reaching the surface layer of the water column is shown in the middle panels for the (**a**) Northern Bering and Chukchi Sea and (**b**) the Barents Sea LMEs for two scenarios SSP2-4.5 and SSP5-8.5. Solid lines show average values with standard deviation in shading. Values are averaged across each LME with a 5-year rolling mean. The dashed lines indicate the relative changes in %-value from the 1980–2000 average (with the scale on the right). The bottom panel shows timeseries of total column ozone (TCO) for SSP2-4.5 and SSP5-8.5.

expected to reach levels equal to around 50 DU (or 10–15%) higher than 1960 values by 2100 (Fig. 4e). Increases under SSP2-4.5 reach values of 20 DU (5–8%). These results are generally consistent with previous estimates of 10–18% decreases in UV radiation under the RCP4.5 and 8.5 scenarios over the 21st century, as derived from simulations from the IGAC/SPARC Chemistry-Climate Model Initiative. Overall, the projected changes in stratospheric ozone will decrease UV-B irradiance for both the Northern Bering and Chukchi and Barents Seas. Still, the amount of UV-B irradiance projected to reach the top of the water column is increasing substantially (Fig. 4, middle panels) by

up to 70% by 2100. This is a consequence of the increased fraction of open water of each LME.

## Changes in chlorophyll-a

Projected chlorophyll-a from CMIP6 models suggests a general increase in biological productivity at high latitudes, but variability across models is high (Supplementary Fig. 1). This variability could be related to differences in model representations of nutrient advection, mixing, and grazing from zooplankton[42]. Our results reflect this variability. Under climate scenario SSP2-4.5, projected CMIP6 chlorophyll-a values suggest increases in annual average chlorophyll-a content of $2.5 \pm 14.5\%$ in the Northern Bering and Chukchi Sea and $11.7 \pm 16.7\%$ in the Barents Sea. We estimated these values as the annual averages from the period 2080–2100 relative to 1980–2000. Conversely, there is a decrease of $3.7 \pm 1.9\%$ in chlorophyll-a content for the Northern Bering and Chukchi Sea and an increase of $7.7 \pm 13.7\%$ in the Barents Sea for the same periods under SSP5-8.5. Overall, we find there is high uncertainty in estimates of future chlorophyll-a concentrations. In contrast, we predict a clear change in the seasonality of chlorophyll-a under both SSP2-4.5 and SSP5-8.5, with the spring bloom estimated to occur earlier for each future decade (Fig. 5, Supplementary Fig. 6). However, the magnitude of change seen in SSP5-8.5 is delayed or lagged by a decade compared to SSP2-4.5. Due to this temporal shift, the Barents Sea chlorophyll-a abundance during May is estimated to increase by 30–50% between 2000 and 2080, while the Northern Bering and Chukchi Sea may experience an increase of 20% by 2080. The Northern Bering and Chukchi and Barents Seas are projected to experience 10–50% reductions in chlorophyll abundance in summer (July–September). The same trends and changes are found under SSP5-8.5, with only small differences (Supplementary Fig. 6).

## Changes in seasonal environmental variability and impacts on pelagic fish

By using laboratory-derived[34] functions for temperature-driven polar cod egg survival (Table 1, Supplementary Fig. 7) and assumptions of i) no acclimation or adaptation and ii) eggs are also in areas without sea ice, we estimate that the projected 1.2–2 °C increase during winter (January–March) in the Barents Sea open surface water temperature may decrease egg hatching success from 85% to 73% by 2100 under SSP2-4.5, or to 63% under SSP5-8.5. Expected temperature changes during winter in the Northern Bering and Chukchi Sea are negligible, and therefore, survival during the egg stage remains unchanged for polar cod as well as for warmer-water walleye pollock (Supplementary Fig. 8). Egg survival within the Barents Sea could increase from 67% to 75% for Atlantic cod (Supplementary Fig. 9) as the duration of the egg stage for that species is shortened.

In contrast to the egg stage, juvenile stages of polar cod grow successfully and survive over a much wider thermal window (−1 to 12 °C) in laboratory studies[43] (Supplementary Fig. 7). However, mortality typically increases with warming waters as aerobic scope, or the ability to perform aerobic activities, declines. The expected increases in temperature during spring (April–June) within the Northern Bering and Chukchi and Barents Seas are within the tolerance limits for juvenile polar cod, suggesting they may experience elevated growth rates given sufficient food (Fig. 5). Juvenile growth rates for walleye pollock in the Northern Bering and Chukchi Sea show that they could also benefit from expected warming, reaching their preferred temperature range by 2050–2060 (Supplementary Fig. 8). In the Barents Sea, warmer conditions will also favor the growth of juvenile Atlantic cod (Supplementary Fig. 9). Hence, all three species may, at least for some period into the future, potentially benefit from warming during the more thermally-flexible juvenile stage.

However, as for all ectotherms, these fish species will increase their metabolic demands with warming and thereby their vulnerability to prey mismatch[39]. Our calculations suggest that prey abundance will

peak earlier (May and April) and have lower levels later in the season (July–September) (Fig. 5) under SSP2-4.5, with even lower levels under SSP5-8.5 (Supplementary Fig. 6). At the same time, expected changes in decadal average ocean temperatures relative to the historical period 1980–2000 suggest that monthly average temperatures will increase by 2.8 °C (0.5 °C) in winter (January) and by 5.1 °C (4.0 °C) during summer (July) by 2100 in the Barents Sea (Northern Bering and Chukchi Sea) under SSP2-4.5 (Fig. 5). These changes are even stronger under SSP5-8.5, with monthly average temperatures increasing by 3.7 °C (1.7 °C) in winter and 6.8 °C (6.9 °C) during summer by 2100 in the Barents Sea (Northern Bering and Chukchi Sea) (Supplementary Fig. S6). The thermal threshold of 4.5 °C for polar cod eggs suggests that expected temperatures will be lethal under SSP5-8.5 by the end of the century (Supplementary Fig. S6).

Further complicating survival are seasonal light changes. Polar cod can visually detect prey when light levels exceed a minimum threshold of $0.1\,\mathrm{Wm^{-2}}$ ($0.457\,\mathrm{\mu mol\ photons\ m^{-2}\ s^{-1}}$) (Laurel pers. comm.), where diminished foraging success appears to occur. This threshold was used as it is in the same range as other cold-water gadid larvae (e.g., Northern Atlantic cod[40]). In the spring, more light due to increased ice-free habitats and subsequently warmer temperatures create conditions favorable for increased larvae and juvenile polar cod growth in both regions. Under SSP2-4.5, the period during winter and spring when PAR exceeds the minimum threshold in the water column will gradually increase from 2.5 to 5 months by 2100 in the Northern Bering and Chukchi Sea, while in the Barents Sea it will increase from 3 to 4.5 months (Fig. 6). The centerpoint of these two large marine ecosystems is 69.5°N for the Northern Bering and Chukchi Sea and 75.0°N for the Barents Sea, which creates differences in their light regimes.

Similarly, the number of months during winter-spring in the Northern Bering and Chukchi Sea when the water temperature is expected to exceed 0 °C will increase from less than 0.5 (average 1980–1990) to between 2 and 5.5 months by 2100, depending on the climate scenario (Fig. 6). This warming is also seen in the Barents Sea where the number of months during winter-spring when the average sea surface temperature exceeds 0 °C increases from 1 month (1980–1990) to 5.5 in 2100 (Fig. 6). Under SSP5-8.5 in the Northern Bering and Chukchi Sea, ocean temperature conditions exceeding minimum thresholds are extended by an additional 2–3 weeks into fall compared to SSP2-4.5, while there is no difference between scenarios by 2100 for the Barents Sea. By contrast, persisting warmer waters in fall will increase metabolic demands for fish, but light levels will remain below those that allow visual predators to feed efficiently (not accounting for moonlight). In fact, for both the Northern Bering and Chukchi and Barents Seas, the summer and fall season that meets the minimum light values needed by polar cod is only extended by an additional week under climate change. Meanwhile, the seasonal duration when ocean temperatures exceed 0 °C continues to increase from 1 (Northern Bering and Chukchi Sea) and 2.5 (Barents Sea) to 5–6 months by 2100 for both regions during summer–fall under both climate scenarios. For winter–spring the number of months that exceed 0 °C increases from 0.5 to 2.5–5.5 months for the Northern Bering and Chukchi Sea and 1 to 5.5 months for the Barents Sea LMEs with the biggest changes occurring under SSP5-8.5. These results indicate an extended period of 2–3 months of elevated temperatures compared to historical values, conditions that will be challenging for all the fish species considered here to accumulate and maintain necessary fat stores for survival through the winter.

## Discussion

Historically, light has been a limiting factor for many ecosystem processes in seasonally ice-covered high latitude regions[4]. However, we find that future increased solar radiation reaching a larger ice-free surface area will raise ocean temperatures and create positive

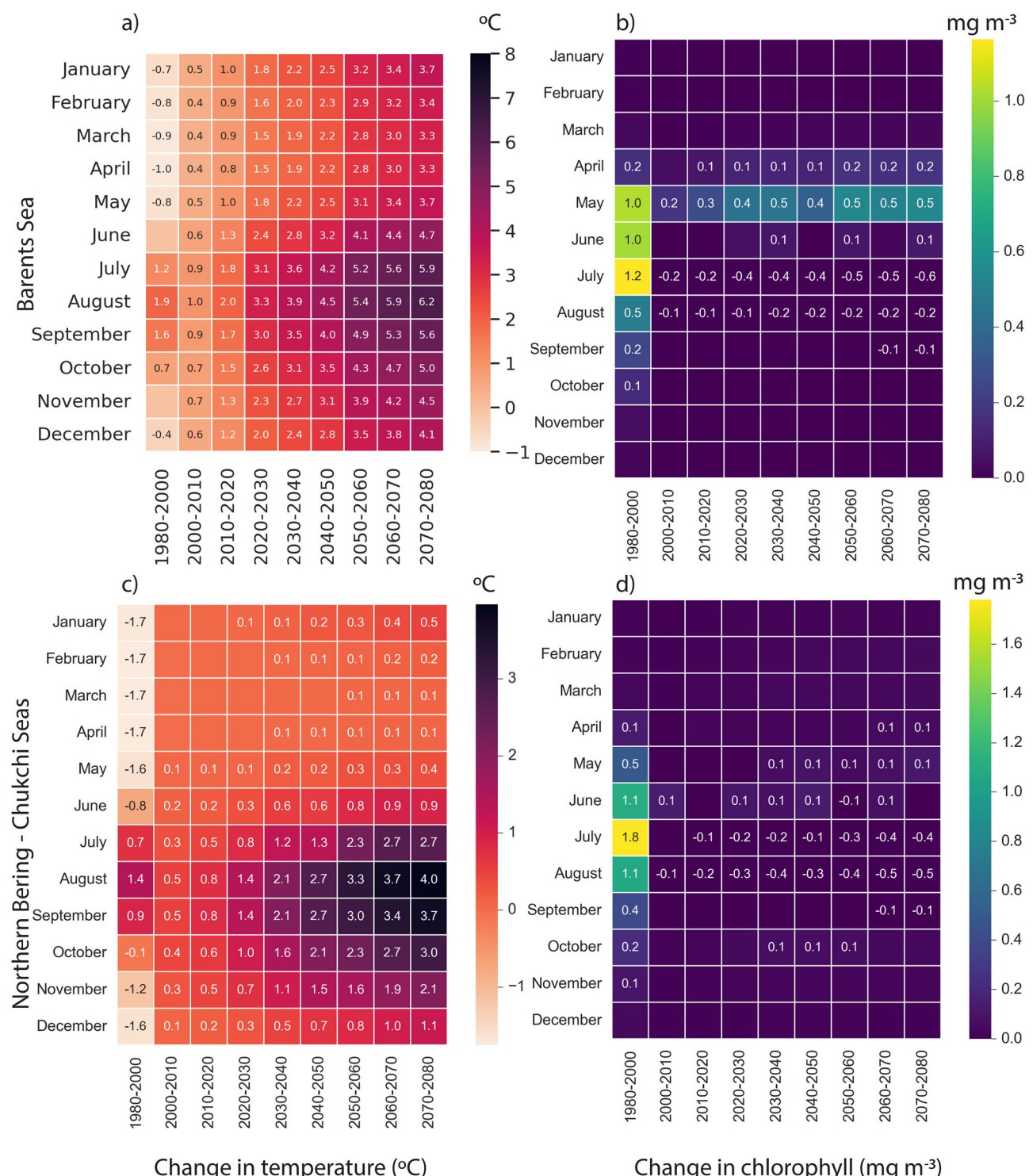

**Fig. 5 | Future decadal changes in temperature and chlorophyll-a abundance.**
Heatmaps show the historical (1980–2000) monthly mean (leftmost column of each heatmap) and future decadal average changes in ocean surface temperature (tos) (°C, left) in (**a**) the Barents Sea and (**b**) Northern Bering and Chukchi Sea and chlorophyll-a (mg/m³, right) for (**c**) Barents and (**d**) Northern Bering and Chukchi Seas under SSP2-4.5. Changes relative to historical values (1980–2000) are shown for each decade between 2000 and 2090 at 10-year intervals. Months with no change relative to historical values have no annotated value.

feedback loops that accelerate ice melt, further warming the water column in a process known as Arctic amplification[44]. Predicted rates of warming may outpace the adaptive capacity of many Arctic fish species[45], particularly those species near the southern limits of their range (e.g., polar cod in the Bering and Barents seas) that are already at the edge of their thermal tolerance limits[46]. Furthermore, elevated

light levels will result in a gradual accumulation of heat during the summer months. As summer concludes and light intensity diminishes, the ocean's heat content may persist at elevated levels for an extended duration. Our results suggest that the combined impacts of reduced snow, thinning sea ice, increased fraction of open water surface area (50–55%, Supplementary Fig. 3), increased light (75–160%, Fig. 3),

increased ocean temperatures (<6.8 °C, Fig. 7), and seasonal shifts in phytoplankton (Fig. 5) in the current light- and nutrient-limited focal ecosystems will cause large changes in the abundance, growth, and survival of key species from phytoplankton to fish. These light-driven changes will create differential rates of survival of fish species, accelerating borealization of the Arctic.

## Reduced egg and juvenile survival for Polar cod

Overall, our results suggest that climate change will create differential egg and juvenile survival rates across two major ecosystems for Arctic and sub-Arctic fish species. For the Northern Bering and Chukchi Sea, winter and spring conditions will likely remain favorable for species such as polar cod. In the Barents Sea, however, warming will negatively impact survival rates of polar cod. These significantly lower survival

rates are potentially due to higher temperatures causing incomplete development of the cardiovascular and other homeostatic systems[47,48] during the egg stage. Polar cod eggs are adapted to the cold under-ice environment, which reduces their exposure to predators and mortality from mechanical stress (e.g., wind and waves). The egg development times for polar cod are 45–90 days, and hatching success is sensitive to environmental changes. The narrow thermal response at the egg stage is typical of many polar species ("stenothermic") and likely reflects tradeoffs between energy efficiency and thermal tolerance at lower temperatures.

It is also possible that faster yolk depletion rates associated with warming will exacerbate match-mismatch consequences for first-feeding polar cod[34,49,50], although this can be true for walleye pollock and Atlantic cod as well. The earlier spring blooms that we predict can cause asynchrony in the timing of larvae hatching and their need to locate prey after yolk-sac depletion to avoid starvation[50]. Conversely, a positive trait of warmer water could be that faster development offsets mortality from predation. At the same time, strong UV-B radiation or prolonged exposure can have non-linear negative impacts on survival and development of fish eggs and larvae residing in the near-surface layer (particularly in the upper 50 cm)[32]. Our results show that the average annual UV-B irradiance that comes through the atmosphere will be less over the coming century due to ozone layer recovery. However, the amount of UV-B that penetrates the surface layer of the water column increases considerably with the fraction of open water. Increased levels of UV-B can cause DNA damage that can be detrimental to egg survival and embryonic development[51,52]. Because of the magnitude of these impacts, warming and changes in irradiance may create a bottleneck in the reproductive success of polar cod in the Barents Sea before they reach the larval and juvenile stages.

**Table 1 | Theoretical optimal temperature for maximum egg survival and juvenile growth rates for Atlantic cod, polar cod, and walleye pollock**

| Species | Egg | | Juvenile | |
|---|---|---|---|---|
| | Optimal temp. (°C) | Survival | Optimal temp. | Growth (%/day) |
| Atlantic cod | 4.3 | 93.0 | 14 | 4.7 |
| Polar cod | 0.2 | 88.0 | 7.5 | 1.5 |
| Walleye pollock | 3.6 | 83 | 13 | 3.0 |

The functional relationships between ocean temperature and egg survival and juvenile growth rates were derived from lab experiments where eggs and juveniles were exposed to a range of temperatures, and their subsequent survival or growth was measured (also see Supplementary Fig. 7).

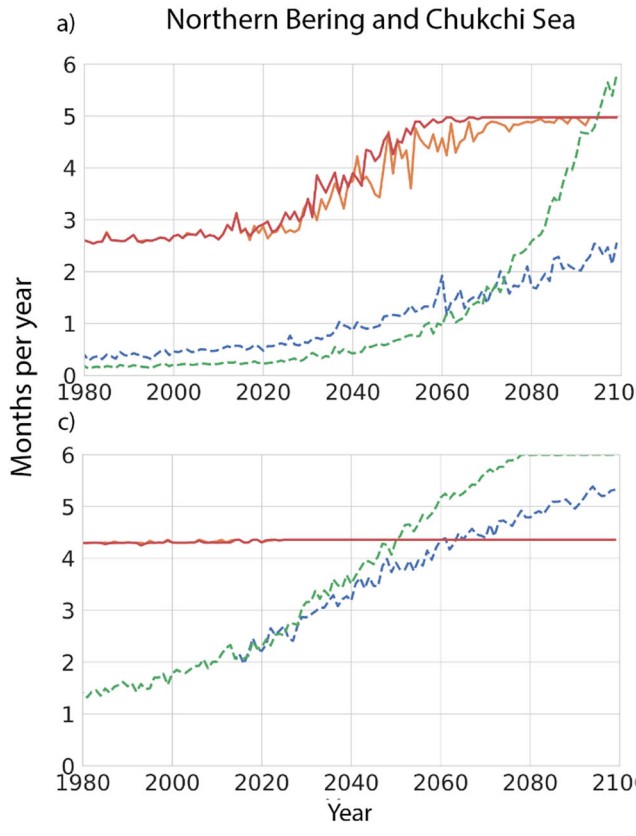
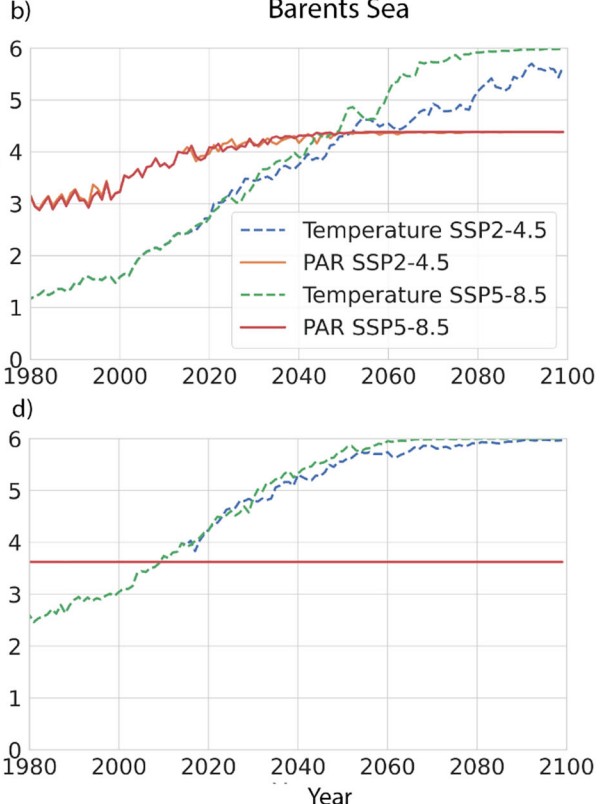
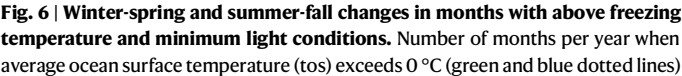

**Fig. 6 | Winter-spring and summer-fall changes in months with above freezing temperature and minimum light conditions.** Number of months per year when average ocean surface temperature (tos) exceeds 0 °C (green and blue dotted lines) and when monthly average PAR in the water column is above 0.1 Wm⁻² (solid red and orange lines) for SSP2-4.5 and SSP5-8.5 for the Northern Bering/Chukchi and the Barents Seas for winter-spring (**a**, **b**) and summer-fall (**c**, **d**).

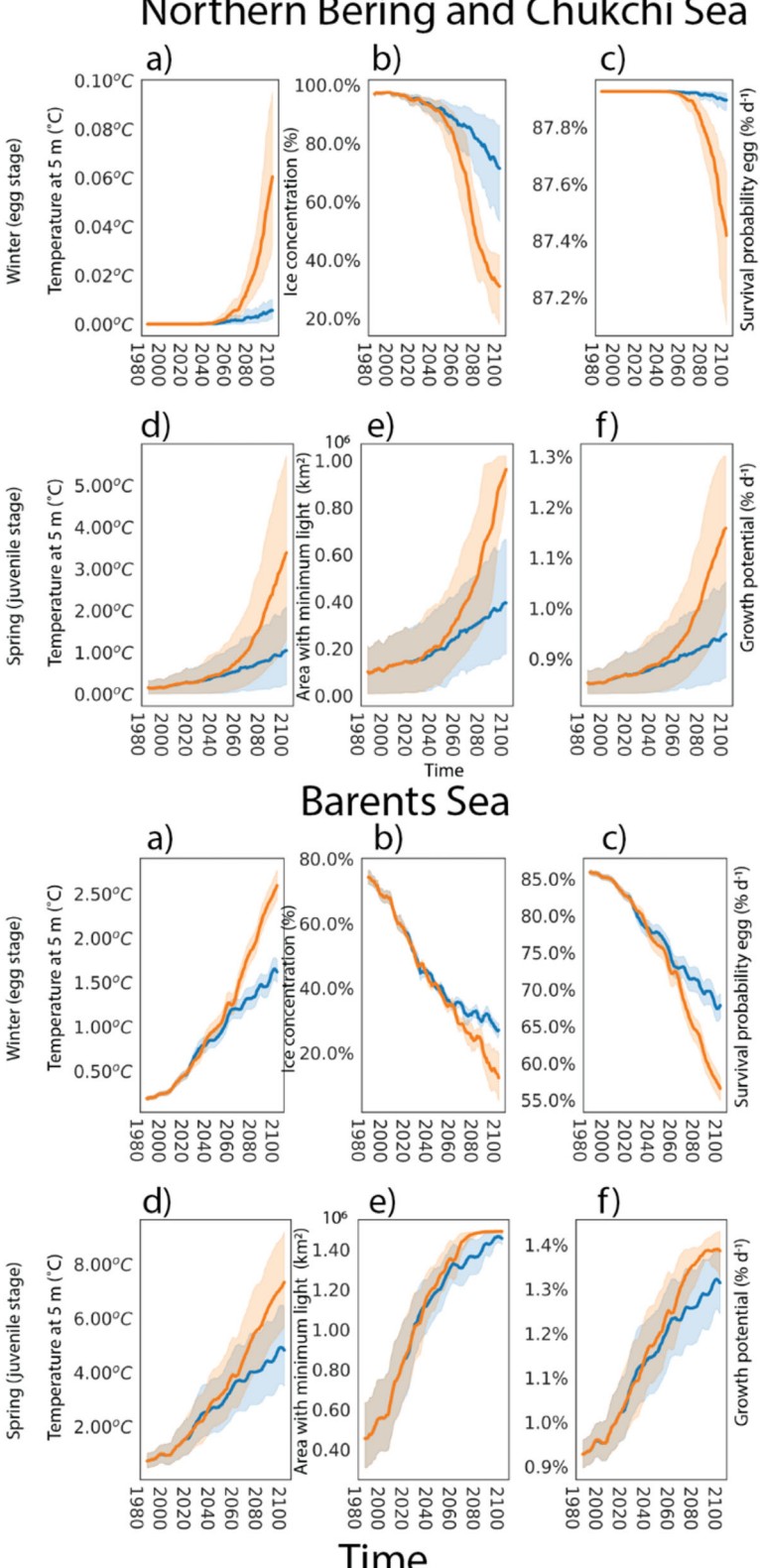

**Fig. 7 | Impacts of temperature, ice concentration, and light on the egg survival and larval growth potential for Polar cod.** Northern Bering and Chukchi Sea (upper) and Barents Sea (lower) panels for winter months (Jan-Mar) and the egg stage are shown in (**a**) ocean temperature (°C) at 5 m depth, (**b**) ice concentration, (**c**) egg survival (% d⁻¹), while summer months for juveniles shown in (**d**) ocean temperature (°C) at 5 m depth, (**e**) area (km²) within the LME with minimum light threshold, and (**f**) juvenile growth potential (% d⁻¹). Thick lines show the average values across all winter/spring months for each scenario SSP2-4.5 (blue) and SSP5-8.5 (orange) while shaded regions show the 95th percentile within the winter/spring months. Survival during the egg stage and growth potential was calculated using functional relationships from laboratory studies and observations[34,43]. Similar figures for Walleye pollock (Northern Bering and Chukchi) can be seen in Supplementary Fig. 8 and Atlantic cod (Barents Sea) in Supplementary Fig. 9.

### Increased likelihood of seasonal mismatches in prey availability

Our findings show that the potential juvenile growth rates of polar cod, walleye pollock, and Atlantic cod are likely to increase with projected temperature rises (Fig. 7). Increased growth rates elevate metabolic demands, which need to be accompanied by adequate prey availability and abundance. However, we find that the timing of open ocean production may shift to earlier in the year (Fig. 5), causing asynchrony with when larval fish will need prey. Earlier peaks in prey abundance will lead to lower prey availability later in the season (July–September, Fig. 7). In the fall, warmer waters will translate to a continued need for food, but food will be harder for visual feeders like polar cod to obtain as seasonal light diminishes (Fig. 6). Although a second bloom could theoretically mitigate these conditions, it is unclear if it will materialize[53] and uncertain how it may impact zooplankton dynamics and abundance[6]. Combined, the increase in temperature and estimated reductions in prey availability are likely to negatively affect polar cod survival during the larvae and juvenile stages, particularly during summer and early fall (Fig. 6).

### Loss of polar cod habitat

Increased light and subsequent changes to phytoplankton and zooplankton productivity will impact the seasonal survival of all three species considered here, but negative effects are strongest for polar cod. In fact, expected changes in ocean conditions at high latitudes suggest polar cod may struggle to find adequate habitats in the Barents or Northern Bering and Chukchi Sea after 2060. We also observe what appears to be a tipping point around 2050 for scenario SSP5-8.5 for the Northern Bering and Chukchi region, where there is a rapid increase in PAR. The increase in PAR follows a strong decline in sea ice concentration and increase in air and ocean temperatures (Supplementary Fig. 1), leading to reduced egg survival and increased juvenile growth potential for polar cod (Fig. 7). The same pattern is also seen for the Barents Sea, although it is less extreme due to the total ice loss potential for this region being much less in 2050 (10%) compared to the Northern Bering and Chukchi Sea (50%). As the fraction of open water increases and open-ocean primary production increases, existing important sources of lipids, such as sea ice diatoms, may decrease as increasing light levels lead to photoinhibitory effects[54].

### Accelerated borealization of major fish species

Boreal-associated species such as Atlantic cod in the Barents Sea and walleye pollock in the Northern Bering and Chukchi Sea may benefit from expected warming and move farther north as both species have wider temperature tolerances compared to polar cod. In fact, observed changes in fish assemblages suggest boreal species are already shifting northwards at a pace reflecting local climate velocities[55,56]. For both boreal species considered here, egg survival and juvenile growth potential increased in response to the expected future ocean warming (Supplementary Figs. 8, 9), facilitating northward distribution shifts. Our results suggest Atlantic cod may expand from the southern Barents Sea into the northern area where polar cod currently dominates. Such northward expansions of Atlantic cod and other species such as herring and capelin, which have similar thermal tolerances, will bring them into polar cod habitats[8]. Similarly, in the Northern Bering and Chukchi Sea, walleye pollock have been observed to expand northwards into polar cod habitats[57]. These shifts are likely to result from warming waters alone, rather than competition displacement because the dietary overlap between polar cod, walleye pollock, and Atlantic cod is moderate[35]. Our regional analyses of key Arctic species suggest that combined changes in light and temperature will increasingly favor first-year survival for walleye pollock and Atlantic cod over resident polar cod in the Northern Bering and Chukchi Sea and Barents Sea.

Climate change may also drive a latitudinal increase in chlorophyll-a production as light and temperature conditions become increasingly favorable at higher latitudes, potentially allowing for the emergence of new spawning grounds for walleye pollock and Atlantic cod farther north. Still, high latitude light limitation during winter may prove to be a constraining factor for poleward full-year range expansion of boreal fish species[11] because they need light to visually detect prey. However, these impacts are still highly uncertain[58]. Fish movements may initially be seasonal, moving northwards during summertime to feed and southwards during winter. This behavior is already exhibited by blue whiting[59], herring, and mackerel[60], which undertake long seasonal feeding migrations and may have range expansions as water continue to warm. Particularly large range expansions could be expected when fish abundances are high, resulting in density-dependent migratory waves[61]. The trophic level consequences of a potentially reduced presence of polar cod are difficult to predict, however, other incoming forage fish like capelin and herring may serve as alternative prey resources for higher trophic level consumers in the Barents Sea, such as ringed seals and seabirds.

In the Northern Bering and Chukchi Sea, the potential for walleye pollock to expand northwards may also be limited to seasonal migrations because of the persistence of winter sea ice, whereas only in the northern Barents Sea may Atlantic cod be limited by seasonal sea ice. Surveys of pelagic fish in the Chukchi Sea between 2012 and 2019 found a drastic increase in walleye pollock (0.1% to 21% of fish abundance)[62] While age-0 group polar cod were substantially more abundant (68–93% of fish abundance), the increased presence of walleye pollock was probably due to increased survival related to warming waters and could be related to the breakdown of the cold pool in the Eastern Bering Sea. The cold-water pool is a subsurface layer of cold bottom water (<2 °C) that forms each winter due to cooling temperatures and brine rejection during sea ice formation[63]. The cold pool acts as a barrier between Arctic and sub-Arctic species, significantly influencing the region's marine ecosystem and species distribution. A recent study found that for the Chukchi Sea, the ocean-atmosphere-ice feedback loop has undergone significant alterations, resulting in persistent warming of this region. This warming has further diminished ice coverage, while enhanced northward transport has increased Pacific-origin waters on the Chukchi shelf during summer months, elevating the transport and abundance of zooplankton[62] essential for the early life stages of fish. Consequently, after the collapse of the cold-water pool in the Bering Sea[64] it may become seasonally viable for fish such as walleye pollock to migrate into new feeding grounds in the Chukchi Sea[62,65].

The effects of more open water, increased light, and higher ocean temperatures, combined with the relatively narrow temperature tolerance of Arctic marine species, suggest that climate change will have major impacts on Arctic marine ecosystems with sizeable differences across ecosystems and seasons. Regional differences will be important when evaluating climate change resilience and adaptation strategies. For the Barents Sea, the winter will be much warmer, with reduced ice cover, while the Northern Bering and Chukchi Sea will be more stable. Spring will occur earlier in both regions, although changes are more rapid and far-reaching in the Barents Sea, leading to reduced survival of egg and juvenile polar cod as well as loss of their potential habitats. While poleward shifts are already occurring, forecasted changes in light regimes will amplify the pace and magnitude of Arctic ecosystem restructuring so that previously boreal species like Atlantic cod and walleye pollock may become increasingly abundant in Arctic waters.

## Methods

### CMIP6 model selection

Four Climate Model Intercomparison Project Phase 6 (CMIP6) models were chosen as input to the radiative transfer model (RTM) calculations based on their evaluated ability to reproduce key aspects of high-latitude dynamics and characteristics. The RTM required input from 12 variables (listed below) from each CMIP6 model across two climate

scenarios. The selected models include six realizations from the Canadian Earth System Model (CanESM5), four and two realizations from the Max Planck Institute Earth System Model, low and high resolution (MPI-ESM1-LR, MPI-ESM1-HR), respectively, and four realizations from the U.K. Community Earth System Model (UKESM1-0-LL) (Supplementary Table 1). For each combination of model and realization, we extracted and concatenated historical data and future projections for two climate scenarios (SSP2-4.5 and SSP5-8.5) to create continuous time series spanning 1979 to 2100. This process ensured consistency in the data across the entire temporal range, allowing for a comprehensive analysis of climate trends over more than a century (Supplementary Fig. 1). Regarding the choice of ensemble members (Supplementary Table 1), for the CanESM5 model we use both 'p1' and 'p2', and for UKESM we use 'f2' rather than the standard 'f1'. For all other datasets, we use the standard configuration 'rXi1p1f1' where only random factors are varied. For CanESM5, the two physical configurations differ in the remapping method for wind fields (bilinear vs. conservative[26]), resulting in minor differences, strongest over Antarctica[26]. Including ensemble members with both 'p1' and 'p2' physics allow us to explore more of the underlying modelling uncertainty. For the UKESM1-0-LL model, the 'f2' configuration is the standard as the 'f1' setting is not used[66].

To minimize uncertainty from model variability, multiple model responses to anthropogenic forcing were weighted according to model performance and independence. The weights were calculated considering the models' performance and independence with respect to multiple observational estimates, including climatology, trend, and standard deviation. Performance weights were calculated using the observational records from NOAA Extended Sea Reconstructed SST v5 (ERSSTv5[67]) and the ocean in-situ dataset Coriolis Ocean database for ReAnalysis (CORA5.2[68]). The CORA5.2 dataset includes temperature and salinity in-situ observations both at the surface and at depth while ERSSTv5 is a global monthly analysis of SST data derived from the International Comprehensive Ocean–Atmosphere Dataset. Weights were calculated using the approach described by Lorenz et al. (2018)[69] where the model performance is accounted for in the enumerator and in the model independence in the denominator:

$$w_i = \frac{e^{\frac{-D_i^2}{\sigma_D^2}}}{1 + \sum_{j \neq i}^{M} e^{\frac{-S_{ij}^2}{\sigma_S^2}}} \quad (1)$$

In Eq. (1), $D_i$ is the distance of model $i$ to observations in space for the three diagnostics (climatological means, trends, standard deviations), $\sigma_D$ defines how strongly performance is weighted, M the number of models and model realizations, $S_{ij}$ the distance between models $i$ and $j$, and $\sigma_S$ defines how strongly model similarity is weighted. The observational ERSSTv5 and CORA5.2 datasets were interpolated to a Cartesian fixed grid of 1×1 longitude-latitude identical to the model grid. Performance metrics were quantified for each diagnostic between modeled and observed sea surface temperature (**tos**) for the high latitudes (66–80°N) and used to quantify the model weights. Sea surface temperature was chosen as it reflects both direct and indirect metric model skill in seasonally sea ice covered waters. Multiple model realizations of the same model were averaged as part of the weight calculations. Overall, a total of 16 models and realizations were used as forcing of the RTM calculations (Supplementary Table 1). The weighted ensemble averages were calculated for each variable (e.g., PAR, UV, UV-B). The weighted ensemble averages were further weighted for calculations involving area averages. The weighted area average for each latitude-longitude cell was calculated as $\delta A = R^2 \delta \varphi \delta \lambda \cos(\varphi)$ where $R$ is the radius of the Earth, $\varphi$ is the latitude, and $\delta \varphi$ and $\delta \lambda$ are the latitudinal and longitudinal spacing between each cell, respectively.

## Calculating spectral irradiance

Irradiance is estimated spectrally using a simple model for the atmospheric radiative transfer of sunlight under clear sky conditions based on specified atmospheric conditions. The spectral band calculations include the wavelengths from 200 to 2700 nm at 10 nm intervals for DNI and DHI, and global horizontal irradiance (GHI) in W m$^{-2}$ nm$^{-1}$ falling on a surface horizontal to the surface of the Earth. Calculations follow the 'simple solar spectral model' for direct and diffuse irradiance on horizontal planes at the Earth's surface implemented as a module in the pvlib[70] library. To account for cloudy sky conditions, we apply a 'cloud opacity factor', where we assume that the radiance of a partly cloudy sky can be estimated as a weighted average of the clear sky and overcast sky. The spectral incoming light was initially determined using the Bird Simple Spectral Model (SPECTRL2[71]). Subsequently, we applied monthly average cloud data (representing cloud coverage per grid cell and available as CMIP6 output) to calculate the relative fraction (rho) of diffuse vs direct sunlight using the Campbell-Norman[70,72] irradiance equations. This fraction was then applied as described by Ernst et al. (2016)[14] to adjust the diffuse and direct spectral light to account for the influence of clouds. The spectral calculations require several input variables. Some were provided directly from CMIP6 model results, such as cloud cover, water vapor content, and ozone thickness. Others, such as albedo, were calculated independently using other CMIP6 outputs. A few variables relied on default parameterizations suggested by the pvlib library[70]. Below, we describe in detail the individual variables and calculations.

## RTM input data

Monthly resolved model outputs from four CMIP6 models and multiple realizations (Supplementary Table 1) were used as input to the light calculations. We calculated the irradiance reaching the ocean surface water (upper 10 cm) under two climate scenarios, SSP2-4.5 and SSP5-8.5, for each combination of CMIP6 model and realization. The RTM, used for the light calculations, required 12 variables extracted from the climate models. The physical and biological model variables included (**variable_id**: description; Fig. 1, Supplementary Figs. 1-2): **tas**: air surface temperature, **siconc**: sea ice concentration; **sisnthick**: sea ice snow thickness, **chl**: chlorophyll content, **sithick**: sea ice thickness, **sisnconc**: sea ice snow concentration, **clt**: cloud cover, **uas**: surface wind velocity east, **vas**: surface wind velocity north, **toz**: total ozone column (measurement of total ozone within the atmospheric column), **tos**: temperature ocean surface, **prw**: precipitable water (integrated water content of the air). All calculations were done using the CMIP6 datasets made available as Zarr archives by the Pangeo community on Google Cloud[73], allowing for calculations without downloading terabytes of data. Prior to calculations, all the required datasets (variables) for each CMIP6 model and ensemble member were interpolated to a Cartesian fixed grid of 1×1 longitude-latitude at monthly temporal resolution using a weighted bilinear algorithm[74]. The RTM was run for each CMIP6 model and realization combination, and scenario independently. The model calculated the light conditions for each grid point (1×1 degree) between 60–85°N and 0–360°E for every 4 h (6 timesteps per day) for the 15th of each month between the years 1979–2100.

## Ozone data

The ozone fields used in the calculations for the PAR and UV-B were obtained from the CMIP6 input4MIPs data archive[75,76]. The dataset was the CMIP-recommended ozone forcing for use in CMIP6 climate model simulations that did not represent atmospheric chemistry interactively. The ozone fields consist of a historical simulation (1850–2014) and different shared socio-economic emissions scenarios for the future (2015–2100). For this study, SSP2-4.5 and SSP5-8.5 were used. Both historical and future ozone fields were generated based on simulations of two chemistry-climate models, the US NCAR WACCM-

CESM and the Canadian ECCC CMAM models, which participated in the SPARC/IGAC Chemistry-Climate Model Initiative Phase-1[75,76].

## Calculating total column ozone (toc)

Ozone strongly affects the amount of light in the UV spectrum that passes through the atmosphere and enters the ocean. This effect is accounted for by scaling the UV light according to a simple relationship between the thickness of the ozone layer and the amount of UV radiation passing through. Not all climate models include an atmospheric chemistry component and therefore do not provide the total column ozone (**toc**) as an output variable. However, most models use the same boundary condition and forcing files input required for the CMIP6 various scenario runs (https://esgf-node.llnl.gov/projects/input4mips/), which enables us to use a common ozone dataset across the models in the calculations shown here. The ozone dataset contains *ozone volume mixing ratios* [mol mol$^{-1}$] for 1950–2100 for each scenario SSP2-4.5 and SSP5-8.5, which was converted to **toz** in Dobson Units (DU) at each grid point (x,y,z) and time (t) for use in spectral irradiance calculations. The conversion was done using the equation:

$$toz = 10 \cdot \frac{(RT_0)}{(g_0 P_0)} \cdot \left( \sum_{i=1}^{N-1} 0.5(VMR(i) + VMR(i+1))(p(i) - p(i+1)) \right) \quad (2)$$

where VMR is the mixing ratio (ppm), N is the number of vertical levels of the air column, R = 287.3 is the specific gas constant for air (J kg$^{-1}$ K$^{-1}$), $T_0$ = 273.15 is temperature (K), $P_0$ = 1.01325e5 is the standard pressure at surface (Pa), $g_0$ = 9.80665 is the global average gravity at the surface (m s$^{-2}$), Na = 6.0220e23 is Avogadro´s number, p is pressure in hPA. Calculated values for the period 1/1/1950 to 12/31/2099 ranged over 218.9–614.7 DU with a mean of 332.1 DU for SSP5-8.5. For scenario SSP2-4.5, values ranged over 218.9–558.8 DU with a mean of 326.6 DU.

## Calculating ocean surface, sea ice, and snow albedos

**Ocean surface albedo.** The albedo over the open ocean is usually a constant equal to 0.06 in climate models and rarely are more complex approaches used. A recent publication[13] proposes a new approach for next-generation climate models that quantifies the effects of the solar zenith angle, ocean waves, and chlorophyll content on ocean surface albedo (OSA), which can help reduce the uncertainty in climate sensitivity to the flow of radiative energy. The proposed OSA scheme, which was implemented in the RTM presented here, calculates the various contributions spectrally from the ocean surface on both direct and diffuse shortwave radiation, providing a more realistic approximation of the reflected shortwave radiation. The complex implementation is described in Séférian et al. (2018)[13] and not repeated here.

**Sea ice and snow albedo.** Sea ice and snow are very efficient mediums for reflecting shortwave radiation and one of the most important factors influencing the Arctic energy budget. However, the efficiency can vary with sea ice thickness, snow crystal structure, and melting ponds, or the purity of ice such as soot particles originating from anthropogenic dust, black carbon from coal combustion, or volcanoes. Here we consider sea ice and snow as pure as we do not have information on the geospatial distribution of any properties that could decrease albedo in the future except purely physical changes. The parameterization of the annual average albedo of thick ($h_{ice}$ > 0.5 m) ice was 0.52 and albedo of snow-covered ice was 0.65. These values are based on the Community Climate Model System Version 3 (CCSM3[77,78]) and the Community Ice Code (CICE[12]) component of the Community Earth System Model and modified to reflect the seasonal changes in sea and snow conditions. The values used for albedo of snow and ice reflect average annual conditions that include seasonal deterioration of sea ice and snow surface conditions away from pure conditions (ice

= 0.73 and snow = 0.96). These values are comparable with recent observed values where average albedo of Arctic sea ice was found to be 0.8 in April–May and decreased to 0.4 between June and August[41,79]. When ice thickness is less than 0.5 m and the ice is not melting, the dry albedo can be defined as $a(dry) = a_o(1 - f_h) + 0.52 f_h$ where $a_o$ is the open ocean albedo and $f_h$ is an asymptotic function defined by $f_h = \min\left( \frac{tan^{-1}(c_{fh}h)}{tan^{-1}(c_{fh}0.5)}, 1.0 \right)$ where $c_{fh} = 5.0$ and $h$ is the ice thickness[80]. To represent impact from melt ponds on the albedo, we used the near-surface air temperature (**tas**) and calculated the wet albedo as $a(wet) = a(dry) - 0.075 f_T$ where $f_T = \min(T_{air} - 1.0, 0.0)$ and $T_{air} > -1.0\,°C$. The algorithms for sea ice and snow can also be found as part of the UCAR Community Earth System Model equivalent to CESM-CICE 5.0 (https://github.com/CICE-Consortium).

## Attenuation from snow, ice, and chlorophyll-a

Earlier observations have shown that the attenuation coefficient through snow can vary considerably, ranging from 4.3 to 40 m$^{-1}$ [18], and ice modeling has reflected this. For example, the CCSM3 model assumed that no shortwave radiation penetrates the snow[80], while the ROMS sea ice module used 20 m$^{-1}$ by default[81]. The latter value was used in this study as it represented a value in the middle of the range of observed values[18], and was also used in Castellani et al. (2022)[4]. Attenuation of snow on top of sea ice was estimated as an exponential decaying function[82] of snow thickness (**snthick**, $h_{snow}$) using a fixed coefficient of $k_{snow} = 20$ [m$^{-1}$]: $I_{ice} = I_{sfc} e^{-k_{snow} h_{snow}}$ for all wavelengths. The spectral absorption coefficients of sea ice describing how light is attenuated as it propagates through the ice was estimated for the wavelength range 200–1000 nm by combining previously published observations[82] that were interpolated to a fixed wavelength of 10 nm. Total attenuation of light from the surface (top of ice, but underneath snow if present) to the underside of the ice was calculated for each wavelength ($k(\lambda)$) as an exponential function[82] of the thickness of sea ice $I_{sfc}(\lambda) = I_{ocn}(\lambda) e^{-k(\lambda) h_{ice}}$. For these calculations, we assumed pure ice, and we did not account for any black carbon (e.g., anthropogenic sources of soot from coal combustion) that would affect the ice optical properties of absorption and reflection. We also do not account for attenuation caused by sea ice algae[83], as the CMIP6 models do not provide that information. We extracted spectral absorption coefficients of phytoplankton (chlorophyll-a) from Table 3 of Matsuoka et al. (2007)[19] and used them to quantify the exponential decay of light with depth due to chlorophyll-a. The chlorophyll in the surface waters results in a further attenuation factor that is independent of the attenuation caused by other seawater components (the water itself, cDOM, other particles). We have assumed that the attenuation from chlorophyll, over a layer (0–5 m), should be independent of the depth distribution within that layer[84].

## Comparison with ERA5 and CMIP6 shortwave radiation

When comparing the monthly averaged global horizontal incoming (GHI) shortwave radiation in our model with ERA5[85] for the period 1979–2020, we found a small, consistent, spatially homogenous bias. This bias can be explained in part by the coarse resolution in our model inputs (1 × 1 degree longitude-latitude) compared to the high resolution of ERA5, and that our RTM model is simpler in capturing the most essential elements, or processes, required to quantify shortwave radiation. Our RTM also lacks several important atmospheric components, such as detailed cloud layering, cloud thickness, and reflection from land, among others. There is also a difference in the land/ocean mask which makes a difference in albedo between the ocean and land between the RTM and ERA5 (not shown). We adjusted for these systematic errors by multiplying GHI by a factor of 1.17. After bias-correcting, the modeled surface incoming shortwave radiation (GHI, 200–2700 nm) had a correlation of r = 0.998 ($p < 0.05$, $n = 480$) and

RMSE = 2.8 W m$^{-2}$ with the monthly mean surface shortwave radiation from ERA5 global reanalysis for the period 1979–2020 (Supplementary Fig. 10).

Next, we compared RTM outputs of absorbed shortwave radiation (incoming shortwave minus effect of albedo) at the surface with those estimated by the individual CMIP6 models (CMIP6 table_ids: **rsds, rsus**) and found that the RTM skillfully replicates the values, ranges, and the dynamical variability between 1979 and 2100. For the Barents Sea LME, the average temporal correlation was r = 0.95 ($p < 0.05$, $n = 60$) and RMSE 4.6 W m$^{-2}$ across 16 CMIP6 model datasets between 1979 and 2100. For the Northern Bering and Chukchi Seas, the correlation was r = 0.91 ($p < 0.05$, $n = 60$) and RMSE = 6.5 W m$^{-2}$ (Supplementary Fig. 12).

The RTM closely aligns with individual CMIP6 models in calculating incoming GHI light and effectively captures the trend and dynamic variability from 1979 to 2100, demonstrating a high correlation (Supplementary Table 2). Particularly, the RTM replicated the estimated light of the CanESM5 and MPI-ESM2-HR models closely, while the UKESM1-0-LL had a higher bias (Supplementary Fig. 13). The RTM effectively replicated trends and relative values across two marine ecosystems with significant seasonal fluctuations in sea ice extent and light conditions, thereby confirming the reliability of our projections of relative changes in light levels.

### Model sensitivity

Some of the RTM model parameters vary considerably when considering the literature, such as the choice of snow attenuation coefficient, which is reported between 4.3 and 40 m$^{-1}$[18]. To understand the importance of some key choices we made for the RTM, and how these affected the final outputs, we performed a sensitivity test. We ran the model for 10 years (1/1/1979 to 1/1/1989) with either a feature turned on or off or a change in parameter value. Features that were turned on or off included the effect of melt ponds (impacts albedo), wind (impacts surface roughness and albedo), chlorophyll (impacts albedo and attenuation), sea ice concentration (impacts attenuation and albedo (not shown in Supplementary Fig. 11)), and the effect of using a snow attenuation coefficient of 5.9 m$^{-1}$ versus the default 20 m$^{-1}$. We also estimated the effect of using the more complex ocean surface albedo (OSA) scheme versus a default value of 0.06 for ocean albedo. We analyzed the average percent difference between the features turned on or off for each individual model and model member in outputs from the RTM over a 10-year simulation (Supplementary Fig. 11). Overall, the mean differences with or without a model feature changed the amount of PAR reaching the water column by <14% for both the Barents and Northern Bering and Chukchi Seas. Change in snow attenuation parameter increased PAR entering the surface waters by an average of 8% (Supplementary Fig. 11) for both LMEs. Removing the effect of chlorophyll (0.05%), and surface wind (1.0–1.16%) also increased the amount of light reaching the surface of the water column by reducing albedo and attenuation. Removing melt ponds reduced the amount of PAR entering the ocean by 0.05% for the Barents Sea and 0.13% for the Northern Bering and Chukchi Sea. The effects from individual features on UV-B were comparable to the effects on PAR, although the relative effect was smaller with changes <7.5%. The effect of implementing a spectral method to calculate ocean surface albedo, OSA, increased the albedo relative to a fixed ocean value of 0.06 with an average impact of reducing PAR by 1.3–1.6% (Supplementary Fig. 11). Generally, the sensitivity tests suggested that changes in our approach or choice of parameter can contribute to changes of up to 15% impact on the annual average amount of PAR that reaches the water column.

### Model limitations

Our analysis only considers the light levels just below the sea surface and does not account for changes in attenuation within the water column, e.g., due to changing concentrations of pigments, colored dissolved organic matter (cDOM), and particulates in part driven by changing terrestrial input. Future work may propagate irradiance down the water column using CMIP model output, which partially covers the previously mentioned variables. However, noting that much of the changes in pigment concentration may occur within subsurface maxima and assuming that most of the newly exposed sea area is not strongly impacted by coastal inputs, it seems likely that the first-order impacts on surface-layer irradiance are already captured by changes in atmospheric attenuation and ice cover/sea-surface albedo as calculated herein. Still, under-ice blooms during spring could impact the attenuation of light reaching the water column, which the model does currently not account for. The effect of clouds, snow, and ice dynamics are all calculated as one dimensional, which reduces the complexity of the RTM, but also means that we are simplifying multi-layered dynamics. A future version of the RTM may include more realistic layering to better resolve such features.

### Biological model

To model the survival of eggs and growth of juveniles of different species, we used functional relationships with temperature as observed in laboratory experiments[34,43,86]. Temperature for the winter months (February, March, April) were used as input to the functional relationship between egg survival and temperature for Atlantic cod, polar cod, and walleye pollock. Summer (May, June, July) temperatures were used to derive the juvenile growth rates for each species. Calculations of either egg survival or juvenile growth were performed for each grid point within the LMEs and then averaged to provide a timeseries with regional uncertainty. The temperatures and values that maximize egg survival and juvenile growth for each species can be seen in Table 1 (optimal temperature). This simple biological model did not consider direct impacts of light on feeding, only the indirect effects of changing light conditions on temperature in the water column.

### Ethics and Inclusion

Our authorship team included one researcher who was based in the Barents Sea region throughout the study design, implementation, data analysis, and partly during the authorship stages. Our results are designed to inform regional rather than local-scale activities, Regional research relevant to this study has been appropriately cited.

### Reporting summary

Further information on research design is available in the Nature Portfolio Reporting Summary linked to this article.

## Data availability

The CMIP6 data used as input to this study are available from Google Cloud Storage (console.cloud.google.com/storage/browser/cmip6).

## Code availability

The source code for the RTM is available on CodeOcean (https://codeocean.com/capsule/675ea292-89f6-456f-ae61-7a6e9ceb7d2b/) and Github (https://github.com/trondkr/RTM, https://doi.org/10.5281/zenodo.15530836), and includes documentation and scripts that allow researchers to create the required forcing files to rerun this study.

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

## Acknowledgements

T.K. was partly funded by the CIMEAS NOAA (704724) grant, the CE2COAST – Downscaling Climate and Ocean Change to Services: Thresholds and Opportunities Project (Norwegian Research Council No. 321890) through the 2019 "Joint Transnational Call on Next Generation Climate Science in Europe for Oceans" initiated by JPI Climate and JPI Oceans, and OSRI grant #22-10-12. ØV was partly funded by the Research Council of Norway through the Nansen Legacy project

(#276730). PW was funded by CE2COAST and the Research Council of Norway through the Migratory Crossroads project (#344079). Computations and data storage were supported by Sigma2 – the National Infrastructure for High Performance Computing and Data Storage in Norway (projects nn8103k, nn9490k, ns9630k).

## Author contributions

T.K. developed the model, did the modeling experiments, interpreted the results, and wrote the manuscript. E.R.S., W.S., Ø.V., and B.L. interpreted the results and wrote the manuscript. MH provided support with the ozone data analysis, interpretation of results, and comments on the manuscript. P.J.W. contributed to the writing, and methods.

## Competing interests

T.K. is associated with the climate data company Actea Inc, which has no role in the content provided here. E.R.S. provides scientific support to companies in the seafood sector through the Seafood Business for Ocean Stewardship (SeaBOS) initiative (https://seabos.org/). None of the SeaBOS members had any role in the study design, analysis, interpretation of data, or conclusions drawn in this paper. The authors have no other competing interests.
