## [Transparent Peer Review file · Nature Communications]

Climate change impacts on ocean light in Arctic ecosystems.

Corresponding Author: Dr Trond Kristiansen

Version 0:

Reviewer comments:

Reviewer #1

(Remarks to the Author)

Please see attached review.

(Remarks on code availability)

The authors made their RTM code available via github.

Reviewer #2

(Remarks to the Author)

I appreciate the authors revising in response to reviewers comments and in particular expanding more on the methodology that was quite vague in the original submission.

My first comment is the authors did not expand the use of the models in their study, just the use of more ensemble members if I understand their response correctly. This is still a weakness. It is still not justified why these four models were chosen and why they didn't use all the CMIP6 models. And in particular, the results will be biased towards the CanESM and the MPI models since more of their ensemble members were used. There is no assessment of how well any of these models reproduce observed parameters going into the RT model. There are publications the authors could refer to and there are plots they could make against satellite observations to support their choice of these models. But in reality we also know that any model may not perhaps get all variables correct in comparison to observations. Already we have become more aware of the albedo schemes in these models being problematic.

Since this paper is about a new RT model, I would find that it would be more appropriate for a paper to be published prior to this one to fully describe the model in more detail than what is done here and to provide sensitivity studies to discuss the most important variables to get right. The authors state a more dynamic sea surface albedo calculation by implementing the impacts of waves and chlorophyll is quite important. But how do we actually know that? It certainly isn't important for under the ice and there is nothing shown in the paper to demonstrate this is critical for your results. I would like at minimum to see the relative contributions of each of the variables to the trends in PAR over time.

Another key issue is these results are computed on a 1x1 degree grid yet snow and ice and albedo, etc. vary a lot on this spatial scale. There should be some sort of distribution function applied to take into consider a more realistic distribution of the key variables impacting light penetration. For example consider a thin snow regime, if you use a skew normal distribution for snow depth, you will not see a large change in PAR because enough light already gets in under that grid cell but for deeper snow regions, continued snow reductions would see an earlier shift of light penetration.

There are a wide range of extinction coefficients one could use for ice and snow and no justification for the values chosen here are given or sensitivity to these chosen coefficients.

I honestly feel this paper should be expanded on to a longer format journal to do the study justice or have another paper first

that describes the methodology in sufficient and justifiable detail before showing future changes, and then use all the CMIP6 models that have the input variables required in order to give a full spread of model and internal variability.

(Remarks on code availability)

Reviewer #3

(Remarks to the Author)

This manuscript titled "Climate change impacts on ocean light in Arctic ecosystems beyond sea ice" by Kristiansen et al. investigates the impact of increased light availability in the Arctic Ocean on the growth and survival of egg and juvenile Arctic and Boreal fish species.

Many concerns raised by the reviewers about a previous version of this draft were addressed by the authors in this revision. I believe this study is a valuable and important contribution to the discussion of possible Arctic ecosystem changes related to increasing underwater light availability. However, I have several comments regarding wording, manuscript structure, applied methods and discussion content, which I would like the authors to address before potential publication. The line numbers relate to the submitted 'clean' draft.

Line 46: Grammar – "and is forced by CMIP6..."

Line 49: Grammar – "while SSP5-8.5 results can be found..."

General note: The manuscript and method description switches between tenses. Please stay in present or past tense.

Line 51: "time of day and year"

Line 52: I think the authors mean "snow-covered sea ice".

Line 53: Better wording could be "as it moves through the snow layer, sea ice,..."

Line 55: More precise wording could be "how light availability is expected to change..."

Line 76: "changing light conditions"

Line 82: This should be worded more clearly "changing light availability" or something similar otherwise it could also refer to light quality, which was not investigated in this study

Line 89: The term "Greater light" sounds very unspecific.

Lines 92 – 93: The "other changes in physical factors simulated in the CMIP6 future model" should be listed here. Because a reduction in snow thickness, increased melt ponds (not melting ponds) and thinning sea ice (no "-" between sea ice) is listed separately in this paragraph (lines 95 – 96), I assume the previous statement about the simulated physical factors in the model does not refer to these physical factors?

Lines 95 – 96: Could you specify the statement about "increased melt ponds"? Do you mean "melt pond coverage"? And does the model output also show an increased melt pond coverage earlier in the year due to increased air temperatures?

Line 98: Will it increase the heat content of the "surface" water column? I'm not sure how much it affects deeper water layers?

Line 99: "positive feedback loop"

Lines 104 – 106: When talking about model output, I think observed changes should be still described a bit more cautiously as predictions. E.g. Instead of describing results as "By 2050, there will be an estimated 25-30% increase in annual average PAR" it should be written as "By 2050, annual estimated PAR is estimated/ predicted to increase by 25-30%...". This applies for several other observations in this manuscript that are worded as definite future truths (e.g. Lines 149 – 150).

Line 113: "as well as it can restrict phytoplankton growth."

Section "UV-B under climate change": This section includes a lot of result interpretation that should be moved to the discussion to make it easier for the reader to focus on the results. Also, references to figures are usually stated at the end of the sentence.

Line 137: Is there any "non-biological" chlorophyll a?

Lines 140 – 142: This sentence should be reworded. It's very hard to read. It should be also worded as full names, Northern Bering/Chukchi Sea and Barents Sea, when % values are given for both. Currently it is stated as Northern Bering/Chukchi without "Sea"

Lines 142 – 143: decrease/increase in which parameter?

It occurs to me that the shown historic negative trends in chlorophyll-a in Figure 6 for both regions do not match the observed historic positive trends in chlorophyll concentration from remote sensing observations (Lewis and Arrigo 2020). Does the biological model include under-ice blooms and lateral biomass advection from other regions that are causing the current observed overall increase in chlorophyll- a in the Arctic Ocean?

Line 153: Omit the dot at the end of the sub-heading

This section about impacts on fish also includes quite a bit of result discussion that could be moved to the discussion section so that it is easier to identify the model results of this study.

Line 184: Missing “under SSP5-8.5”

Lines 184 – 187: If values in brackets are referring to the Northern Bering/ Chukchi Sea, then the months describing the seasons should not be given in brackets as well. Otherwise, it is very confusing.

Line 195: “ $\mu\text{mol photons m}^{-2} \text{ s}^{-1}$ ”

Lines 201 – 216: It is very difficult to connect this paragraph with the previous paragraphs because it’s suddenly a mix of statements about the PAR threshold and the temperature threshold. I think this paragraphs needs a bit more structure, so that it is possible to follow the story.

Line 221: It’s better to say “positive feedback loops”

Line 222: Maybe reword to “making it more difficult for long-lived marine species to adapt”.

Line 223: “increased light levels”

Lines 225 – 227: Although this statement about a mismatch between energy requirements and light availability for visual feeding is further explained three paragraphs down, it seems to not be well connected in this section. I think the flow of this discussion section could benefit from restructuring to connect the statements in this paragraph and the next with the explanations in the paragraphs below.

Lines 234 – 235: This sentence repeats the information of the previous sentence and is not needed.

Line 250: Changes to who’s productivity? Phytoplankton, zooplankton, fish?

Lines 253 – 254: Earlier peaks in prey abundance due to earlier phytoplankton blooms? What impact do fall phytoplankton blooms, which are not uncommon in the Barents and Bering Sea (Ardyna et al. 2014), have on prey availability?

Lines 256 – 258: How does the decrease in ice cover and increase in UV affect Polar cod egg survival?

Lines 280 – 283: The statement about Polar cod seems to not fit well into this sentence. This could be a separate sentence that also provides percentages (or change in percentages) of polar cod abundance from these surveys for more context.

Line 285: The term “cold-water pool” should be explained more.

Line 291 “for a poleward full-year range expansion...”

Although changes in chlorophyll-a from the model output are presented in the result section, they are surprisingly not mentioned in the discussion. Could the authors discuss how the predicted changes might impact prey availability and connected to that the abundance, growth and survival of key fish species?

The presented predictions of changes in UV-B that reaches the ocean are also not mentioned in the discussion. However, as highlighted above, the impact of UV-B on fish species is discussed in the results section, which I think should be moved into this section to provide a more in-depth discussion on how changes in ocean light may affect fish.

Lines 308 – 311: Could you break this very long and cumbersome sentence into two sentence, which will make it easier to read. Also, what were the 11 required variables mentioned here?

Line 309: Could you define “RTM” here again?

Line 320: “from one to another”

Line 330: “and in the model independence...”

Line 356: Where was the monthly average cloud data downloaded from or was it simulated? Please provide more information.

Line 357: Does the correction factor reference to the rho fraction?

Lines 406 – 408: The sentence currently reads “Not all climate models [...] do not provide the total column ozone...”. Is this

the correct wording?

Line 438: "melt ponds"

Line 443: The value for fh is not provided.

Line 443: "To represent the impact of melt ponds on albedo,...."

Lines 443 – 447: How was the change in melt pond coverage calculated in the models?

Line 456: I think that the assumption of pure ice could introduce a considerable error in calculated under-ice light availability in spring when the ice algae layer can strongly absorb light within the ice layer, especially in the blue and red spectrum (Ehn and Mundy 2013, Perovich 1996). Given the patchiness and variability in ice algal biomass, the changes in snow depth probably have a much larger impact on changes in light transmission on a pan-Arctic Scale. However, I think the reasoning for excluding a chlorophyll layer in the ice cover in spring should be briefly mentioned here.

Line 486: "and UV was integrated from 280-400nm."

Lines 496 – 500: I disagree that changes in pigment concentration will only have a small impact on light availability in the water column. I agree that subsurface maxima are a common feature in the Arctic summer months. But this manuscript highlights that light availability will particularly increase in the spring and fall months due to the thinning and disappearing sea ice cover. During these periods, large under-ice blooms, especially in the Chukchi Sea, and fall blooms, a common feature in the Barents Sea, can have a large impact on light availability and visibility for visual hunters.

Figures

Figure 1: Currently it looks like as much light is transmitted through bare ice as it is through thick snow-covered ice. That's not really true (the transmitted arrows are equally thick). I also think it would be more accurate to call the green cells "phytoplankton" instead of just "chlorophyll".

Figure 3: Please define PAR in the figure caption again (photosynthetically active radiation). Also, the results section does not explain the sudden significant increase in PAR after 2050 for the SSP5-8.5 scenario in the Northern Bering and Chukchi Sea. Why is there such a large increase? When combining this plot with the content of the supplementary figures, the year 2050 seems to generally be a tipping point in SSP5-8.5. Can this be explained more in the manuscript?

Figure 5: Impacts of temperature, ice concentration and light on egg survival and larval growth potential; not just temperature and light according to the y-axes. The figure caption has some grammatical issues in the description of what is shown on each y-axis. Additionally, the y-axes of the middle column of sub-plots are very cramped in.

Figure 6: I recommend to label the x-axes "Change in Temperature" and "Change in Chlorophyll"

Line 758: "are shown"

Figure 7: The axis labels are very small, hard to read. And the figure caption does not describe what the solid and dotted lines are showing.

Line 765: "monthly average PAR"

Supplemental figures:

Figure S1: Chlorophyll subplot – The y-axis unit "1e-7" is much too small. Also please label it Chlorophyll-a

Figure S2: The term "predictable water" is uncommon. What does it mean?

Figure S3: Axis labels are too small. Also, the resolution of this plot is very low. When zooming in to better read the lines and labels, the plot becomes very blurry.

Figure S4: The term "PAR" should be used instead of "Ocean light" to be more clear about the wavelength range and to match the plot label.

Figure S5: Some verbs are missing in the sentences of the figure caption.

Figure S6: The axis labels are too small again. And the citation lists "upper panels", but there are no other panels in this plot.

Figure S7: How is this figure different from Figure 6 in the manuscript?

Figure S8: "Functions are shown for a)..."

(Remarks on code availability)

Version 1:

Reviewer comments:

Reviewer #1

(Remarks to the Author)

I thank the authors for spending time reviewing their manuscript based on the comments from all reviewers.

A much needed sensitivity analysis has been added as well as comparison of the model outputs to reanalysis. This helps the reader assess the model uncertainty.

I believe that all comments from reviewer 1 and 3 have been addressed. However, I still express some doubts as to how the authors justify some of their decisions (i.e., limiting runs because files become too large...).

Regarding the remarks from reviewer 2, I acknowledge that the authors have tried to answer concerns or to provide answers when they disagreed with the comments. The authors justified their use of a limited number of CMIP6 models which is fine. I struggle to understand their choice of ensemble members across models though. I believe that the authors justified themselves by saying that "By utilizing multiple ensemble members across various climate scenarios, we aim to capture a broad range of potential outcomes." which is true but by selecting some ensemble members over others without truly justifying it, they introduce a form of bias to their results. Most of my worries however, reside in the fact that aside from using different realisations between models, the authors also used different forcing and physics settings (see Table S1) which are not consistent between models. How can results therefore be analysed together or compared ? Are the authors aware of this ?

A minor detail but I would still suggest that the authors carefully proof read their manuscript as I noticed a few typos here and there.

(Remarks on code availability)

Version 2:

Reviewer comments:

Reviewer #1

(Remarks to the Author)

All comments have been addressed by the authors. Thank you for taking on board the suggestions of the reviewers. In our view, the manuscript can now be accepted for publication.

(Remarks on code availability)

January 23 2025

Reviewer #1 (Remarks to the Author):

Review of manuscript entitled "Climate change impacts on ocean light in Arctic ecosystems" by Kristiansen et al.

Having reviewed a different version of this manuscript before, I thank the authors for carefully considering previous remarks in this new version they submitted. The manuscript is still focused on the use of a newly develop method to estimate the propagation of solar radiation from the top of the atmosphere to the bottom of sea ice, how the light availability and warming might change in the future and the impact on cold-water fish species.

I appreciate the work done by the authors to expand their studies to include more CMIP6 models and realisations to offer the reader a broader understanding of the inherent variability between each model and their variants. The study is, I think, interesting but some details are still lacking in my opinion to have it ready to be published.

We appreciate the reviewer's kind words. The reviewers' comments and suggestions were very helpful, and we have made an effort to address all reviewers concerns in this major revision.

I would say that the one of the primary issues resides in a proper parameterisation of the light attenuation by snow and sea ice despite the emphasis of the study on future changes in snow and sea ice condition (see comment for this paragraph). Secondly, as this is a modelling study, I think it would benefit from more comparison to in situ datasets regarding actual levels of light and parameterisation, this is very much lacking now.

We have now conducted a comparison with ERA5 reanalysis on this latest version of the RTM where a few changes were made to the model (e.g., changes in sea ice and snow albedo). We have also added an extensive comparison of the RTM outputs with the output of irradiance as estimated by individual CMIP6 models to quantify how well our model is able to reproduce outputs from the climate models. Finally, we have added a detailed sensitivity analysis of the RTM that adds a lot of information as to what elements of the model are particularly sensitive to changes (e.g., snow attenuation). We hope that the updated and added elements of comparisons and sensitivity tests are informative and satisfy the reviewers' concerns.

I'm also still a bit puzzled by the RTM. The authors refer to it as a "newly developed RTM" and now properly explain the different parts in the Method section, but it appears that this RTM is simply different published algorithms assembled. The concept of a RTM from toa to bottom of sea ice is interesting and I acknowledge that there is considerable work behind this study, but, in my opinion, with the focus of this manuscript on change in snow and sea ice conditions, a stronger emphasis should have been put on the modelling of light transmission through snow and sea ice and this part is underwhelming.

We agree with the reviewer that this wording was misleading and now describe our methodology used more carefully. One reason we created the RTM was to understand how light at various wavelengths reaching the ocean is affected by changes in the atmosphere, cryosphere, and ocean conditions under climate change across climate models and scenarios. Shortwave radiation is available from the individual CMIP6 models, but only as integrated irradiance, which prevents you from diving into the changes of the various components of the spectra (e.g., UV, PAR). The RTM we present solves this problem and allows the user to calculate spectral light at the air/sea interface and at depth in the ocean. This is done by using 12 variables from the CMIP6 models as forcing for the RTM. The RTM provides a simple modeling framework that can be used by researchers to run different light experiments to analyze changes in light in the ocean, rather than having to run full complex climate models. As the reviewer comments on, the RTM consists of elements from individually published algorithms and certain elements of the model are simplified, such as the one-dimensional atmosphere and one-layer ice and snow dynamics. Still, the RTM is a standalone model that allows you to explore changes in the light regime easily without having to tackle the intricate CMIP6 models which consists of individual modules for the atmosphere, sea ice, ocean and land that are interconnected and cannot be run independently. Our RTM solves that problem and as can be seen in the added comparisons and

analysis, we can correctly reproduce the inter- and annual variability in shortwave radiation as projected by the individual CMIP6 models with our RTM. In fact, our RTM correctly captures the trends and variability in future shortwave radiation as projected by the CMIP6 models for each LME of interest extremely well ($r=0.95$, $RMSE=4.6W/m^2$ for the Barents Sea LME) suggesting that the RTM produces reliable results even if some of the elements of the RTM are simplified compared to a full climate model. This gives us confidence in the results of our RTM.

Please see my various comments below:

Line 30-31: I'm not sure I agree with this sentence. More and more studies have been published over the years that are focused on the response of ecosystems to change in light availability in the Arctic, sure it's still a challenging topic but lots of progress has been done, see Tedesco et al., 2019; Connan-McGinty et al., 2022 or the recent Flores et al., 2023 which it seems you might have not seen.

We have added some more references to the sentence on light and changed our wording slightly. Thank you for the paper suggestions - I had not seen the Flores paper which was extremely interesting, and which is now referenced in the manuscript.

Line 41-45. Is it newly developed because it combines various algorithms that were not associated before ?

Yes correct. We have reworded these sentences and we now say: "We quantify how climate change will affect light regimes in the Arctic using a new approach that allows for the detailed analysis of the large-scale spectral changes in shortwave radiation under climate change. To this end we combine individually published RTM algorithms that are used to quantify the spectral albedo from waves and...."

Line 160: correct SSSP2-4.5

Fixed

Line 180: I think it would be very interesting to compare your results to Flores et al 2023. They did a similar study for the whole Arctic looking at light levels for zooplankton vertical migration from an observation point of view but also by using CMIP6 future projections.

I agree a comparison would be interesting, but I think outside the scope of this manuscript. I did look at the code they used for calculating light and I expect we would get similar trends if we compared although there are some differences in approach. This could be a follow-up project or potential collaboration for a proposal.

Line 195. Interesting, first time I see you now use the proper unit for PAR when used for biological purposes. Although I do understand that you want to reach a broader audience by using the physical unit $W m^{-2}$ for PAR, the work in this manuscript will be of greater interest for scientists interested in biological outcomes of increase in light availability and they are usually more used to the proper unit (i.e. you had to use it in parenthesis at some point to compare to other studies).

I agree with you but to make units easy to understand we decided to keep the W/m^2 .

Line 307.

Regarding models, I would like to bring your attention to the sensitivity study from Dirk Notz on Arctic sea ice within CMIP6 (Notz 2020) where a certain number of CMIP6 models were found to properly simulate sea ice loss while also properly representing global temperature increase over time. Two of the models you used are in the list but not the other twos.

Our choice of models were based on how well the models represented the climate dynamics of seasonally sea-ice covered seas in peer-review literature, and the availability of a range of variables across multiple climate scenarios (not all climate models provide all variables). Our methods weighted the skill and independence of each individual model to the ensemble, providing a robust approach for combining a range of climate to estimate the overall physical impacts on light conditions at high latitudes. We do reference the excellent Notz paper in the introduction.

Line 346. I understand that you used several CMIP6 variables such as cloud cover and ozone layer combined with the amount of solar radiation reaching the top of the atmosphere to simulate the

proportion of sunlight reaching the snow/sea ice/ ocean surface. Did you try to consider the downwelling shortwave at the surface variable from these same models and compare with your method ? and if not, why not ?

We appreciate the reviewers' comment which triggered us to add an additional analysis to the manuscript. This major revision includes a direct comparison of our RTM and the individual CMIP6 models at the ice/ocean interface. We believe the added figures (Figs. S12 and S13), table (Table S2) and discussion (see section "Comparison with ERA5 and CMIP6 shortwave radiation") adds important information and validation of the RTM and we appreciate the reviewer's request.

Line 380-382. I find it awkward to use monthly (and not daily) outputs but then calculate every 4 hours for the middle of the month. Light is such a sensitive variable to daily processes. What is the reasoning behind all this ? Is it because it might be computationally too heavy ? Please explain.

The model is computationally heavy to run for multiple climate models and scenarios for the period 1979-2100. For efficiency, we run the model one day per month, but to resolve the daily light cycle we did the calculations at 4-hour intervals. We also ran the model using 1 hour resolution and compared the outputs and found the differences to be negligible, but the time of execution increased considerably. Hourly resolution gives you slightly smoother daily graphs of the hourly values, but the values are the approximately same as the values (e.g., 12 to 16 pm) are linearly interpolated when we calculate the ensemble values. The size of the output files also grows considerably from tens of gigabytes to hundreds.

Line 385-391 This whole paragraph is a repetition of a previous paragraph. There is quite some repeating in the method section, you might want to avoid that.

We agree and have removed the sentences from the manuscript.

Line 435-438. There is no problem in using pure snow or sea ice properties, however this sentence (here and elsewhere) put the emphasis on the impact of soot particles which you are not considering. If you desire to keep it like that, you should add a few lines in the discussion about the impact of not considering soot particles.

We have rewritten the sentence.

Line 438-439. Why not look for measured albedo values instead of modelled ones ? There are so many in situ studies on albedo of Arctic sea ice. You'd then see that an albedo of 0.96 for snow covered sea ice happens with (very) fresh snow and it is rarely seen in situ. It is even less seen in a melting Arctic (i.e., Perovich and Polashenski 2012). Albedo reduces because of melt pond of course and it is good that you account for that, but it also reduces because of snow metamorphism over time. Please explain.

We completely agree and as you will see in this version of the manuscript, we have revised our values for sea ice and snow albedo. This came about after direct comparison between model and observations. We now base our values on what we find in literature and observations, and we account for the seasonal variability of the albedo values. Our estimated average values are comparable with what we find in the literature.

Line 443. Missing value: $F_h = ?$

Fixed and we added the equation for f_h .

Line 453-458. I'm kind of surprised regarding the attenuation through snow and sea ice. Several studies covering in situ and remote sensing data as well as model based have been using the exponential decay relationship to evaluate solar partitioning by snow and sea ice before, this is not new, and it seems what you've used is even less complex than these studies (i.e., Light et al., 2008, 2015, Lebrun et al., 2023, Stroeve et al., 2021) as not considering a drained layer in sea ice or even the existence of a surface scattering layer forming due to interaction of solar irradiance with the top layer of sea ice. It is a bit underwhelming considering that the study is focused on the change in snow and sea ice conditions on under sea ice light availability.

We agree that our one-dimensional approach has the potential to be improved in the future. Our focus was on creating a model that was realistic and fast, but there is room for improvement including a drained layer in sea ice.

Regarding the extinction coefficient in sea ice: it is mentioned that it's a spectral extinction coefficient but no mention of the values nor where it is coming from ? Regarding the extinction coefficient in snow: I am very surprised of the value used of 0.2 m^{-1} . It is extremely small, even way lower than the usual broadband extinction coefficient for sea ice commonly used.

It is even more curious considering the albedo of 0.96 that you used. Where is this value originating from, is it for a single wavelength ? It is a broadband value as well which makes it even more confusing that it's so low. Please explain the reasoning and references behind this.

I would suggest you have a look at the literature for extinction coefficients in snow (i.e Perovich 1996 or the recent paper by Lebrun et al., 2023). I would also suggest considering different conditions for snow evolution stages where albedo and extinction coefficient change accordingly.

We apologize but the value of 0.2 m^{-1} was a typo. The correct value is 20 m^{-1} . We have updated the text with references as to where this value originated from, and we have also added a sensitivity test where we test two different values of the attenuation to see the impacts on PAR into the ocean. We hope this correction makes sense to the reviewer and resolves this misunderstanding. We do not account for snow evolution stages (currently) but rather use a single value for sea ice and snow albedo that reflects the seasonal variability through its mean value.

Reviewer #1 (Remarks on code availability):

The authors made their RTM code available via github.

We have updated our source code on Github and will also distribute the code through Nature Portfolio.

Reviewer #2 (Remarks to the Author):

I appreciate the authors revising in response to reviewers comments and in particular expanding more on the methodology that was quite vague in the original submission.

My first comment is the authors did not expand the use of the models in their study, just the use of more ensemble members if I understand their response correctly. This is still a weakness. It is still not justified why these four models were chosen and why they didn't use all the CMIP6 models. And in particular, the results will be biased towards the CanESM and the MPI models since more of their ensemble members were used. There is no assessment of how well any of these models reproduce observed parameters going into the RT model. There are publications the authors could refer to and there are plots they could make against satellite observations to support their choice of these models. But in reality we also know that any model may not perhaps get all variables correct in comparison to observations. Already we have become more aware of the albedo schemes in these models being problematic.

We appreciate the reviewer's comments regarding our model selection and weighting approach. As previously explained, we have carefully weighted the models based on their independence and frequency. For instance, we have appropriately adjusted the weighting for the CanESM model due to its higher representation in our ensemble. We respectfully disagree that not using all CMIP6 models diminishes the robustness of our study. In fact, our analysis incorporates a larger number of CMIP6 models and ensemble members than many comparable publications, with the exception of those focusing on a single variable such as sea surface temperature. It's worth noting that it's not standard practice to require the use of all existing CMIP6 models for climate projection studies, as this would impose an unreasonable burden on researchers. This would be akin to expecting all studies using ocean reanalysis products to incorporate every available reanalysis. We believe that our selected models represent a well-established set of climate models that have undergone thorough validation and testing by experts in recent years. By utilizing multiple ensemble members across various climate scenarios, we aim to capture a broad range of potential outcomes. Our approach of running the RTM with each model and ensemble member combination allows for a comprehensive analysis of individual model contributions and helps elucidate the factors driving the spread across model solutions. In our view, our approach presents a robust and diverse selection of climate models, and we believe that the

uncertainty levels in our results accurately reflect the general underlying model uncertainty. To address concerns about the RTM's parameterization and feature effects, we have conducted sensitivity tests. We hope this test provides valuable insights into the RTM's sensitivity to certain parameterizations, such as the impact of snow attenuation on PAR. We sincerely thank the reviewer for their thoughtful feedback, which has helped us strengthen and clarify our approach.

When it comes to validation of the independent CMIP6 model output, we have now added an extensive comparison between the RTM and the individual CMIP6 models. We directly compare how well the RTM can replicate each model and we have provided several new figures showing the overall comparison across all CMIP6 models and the RTM outputs (Fig. S12) and individually (Fig. S13) and the overall summary of the temporal correlation between 1979-2100 and the average RMSE (Table S2). We have also updated our comparison between the RTM and ERA5 reanalysis for the period 1979-2020. We hope these additions satisfy the reviewer's request.

Since this paper is about a new RT model, I would find that it would be more appropriate for a paper to be published prior to this one to fully describe the model in more detail than what is done here and to provide sensitivity studies to discuss the most important variables to get right. The authors state a more dynamic sea surface albedo calculation by implementing the impacts of waves and chlorophyll is quite important. But how do we actually know that? It certainly isn't important for under the ice and there is nothing shown in the paper to demonstrate this is critical for your results. I would like at minimum to see the relative contributions of each of the variables to the trends in PAR over time.

We agree that a sensitivity analysis of some of the underlying features of the RTM that we claim are important was needed. As mentioned above, we have conducted a sensitivity test of how important select variables and features of the RTM are on outcome variables. For example, based on comments from reviewers we tested how important parameterization of the attenuation coefficient for snow is. We tested using two values of 5.9 and 20 m^{-1} and we quantified the percentage change of several variables including total PAR and UV light into the ocean for each value. We also tested how important the OSA (ocean surface albedo), melt ponds, the effect of chlorophyll, surface wind (surface roughness) features of the RTM were on various outputs (Fig. S11). In general, we find that the impacts of these features were small except for snow attenuation (4-14% effect on PAR). Please see section "Model sensitivity" for more details.

Another key issue is these results are computed on a 1x1 degree grid yet snow and ice and albedo, etc. vary a lot on this spatial scale. There should be some sort of distribution function applied to take into consider a more realistic distribution of the key variables impacting light penetration. For example consider a thin snow regime, if you use a skew normal distribution for snow depth, you will not see a large change in PAR because enough light already gets in under that grid cell but for deeper snow regions, continued snow reductions would see an earlier shift of light penetration.

Our calculations are done on the same resolution as the CMIP6 model outputs. The impact of snow depth on light in a particular cell accounts for the snow distribution within that cell. There is no way for us to account for a higher degree of spatial dynamics within that cell, but our approach is identical to the way these features are calculated within climate models. Naturally, we agree the real world has a much higher degree of spatial variability and our estimates have to be considered more of a spatial average value.

There are a wide range of extinction coefficients one could use for ice and snow and no justification for the values chosen here are given or sensitivity to these chosen coefficients.

The choice of spectral attenuation values through sea_ice was done using published values we extracted from the publications and interpolated to a 10 nm frequency (described in methods). We used published values extracted from the publications of Warren et al. 2006 and Perovich et al. 1996 to determine the choice of spectral attenuation values through sea_ice, which we then interpolated to

a 10 nm frequency (as described in methods). There are limited publications of spectral attenuation through sea_ice available, and we believe our choice makes sense.

For attenuation through snow we agree that the choice of value can be quite variable ranging from 0-40 m⁻¹. In the CCSM3 model it assumed that no shortwave radiation penetrates the snow, i.e. $k_s = \text{Inf}$ (see p25 in Briegleb et al. 2004). A value of 20 m⁻¹ was used in the original (Budgell) ROMS sea ice module. This is the value we have used for our RTM as it represents a value in the middle of the range of observed values 4.3 – 40 m⁻¹ (Perovich, 1996). It was also used as a higher test value in Castellani et al. (2022).

We have tested the implications of the choice of this value on the RTM model output and we compared with the effect of using a value of 5.9 m⁻¹ which is based on observational data fitting (Lebrun et al., 2023). We agree that our choice of attenuation coefficient for snow should be better argued for and have added an explanatory section to the methods.

I honestly feel this paper should be expanded on to a longer format journal to do the study justice or have another paper first that describes the methodology in sufficient and justifiable detail before showing future changes, and then use all the CMIP6 models that have the input variables required in order to give a full spread of model and internal variability.

We believe that the article including both the light changes under climate change and the relation to biology makes this article different from just a methods paper or a biology paper but integrates and discusses how physics affects biology.

Reviewer #3 (Remarks to the Author):

This manuscript titled "Climate change impacts on ocean light in Arctic ecosystems beyond sea ice" by Kristiansen et al. investigates the impact of increased light availability in the Arctic Ocean on the growth and survival of egg and juvenile Arctic and Boreal fish species.

Many concerns raised by the reviewers about a previous version of this draft were addressed by the authors in this revision. I believe this study is a valuable and important contribution to the discussion of possible Arctic ecosystem changes related to increasing underwater light availability. However, I have several comments regarding wording, manuscript structure, applies methods and discussion content, which I would like the authors to address before potential publication. The line numbers relate to the submitted 'clean' draft.

Line 46: Grammar – "and is forced by CMIP6..."

Corrected

Line 49: Grammar – "while SSP5-8.5 results can be found..."

Corrected

General note: The manuscript and method description switches between tenses. Please stay in present or past tense.

Line 51: "time of day and year"

Corrected

Line 52: I think the authors mean "snow-covered sea ice".

Corrected

Line 53: Better wording could be "as it moves though the snow layer, sea ice,..."

Corrected

Line 55: More precise wording could be "how light availability is expected to change...."

Corrected

Line 76: "changing light conditions"

Corrected

Light 82: This should be worded more clearly "changing light availability" or something similar otherwise it could also refer to light quality, which was not investigated in this study

Corrected

Line 89: The term "Greater light" sounds very unspecific.

Changed to "Enhanced light availability in the water column"

Lines 92 – 93: The "other changes in physical factors simulated in the CMIP6 future model" should be listed here. Because a reduction in snow thickness, increased melt ponds (not melting ponds) and thinning sea ice (no "-" between sea ice) is listed separately in this paragraph (lines 95 – 96), I assume the previous statement about the simulated physical factors in the model does not refer to these physical factors?

The reviewer assumes correct. We are referring to the atmospheric forcing and have listed these now. The sentence reads: "In response to the increased fraction of open water, mainly driven by reduced sea ice concentration (Fig. 2, S1-2), as well as other changes in physical factors simulated in the CMIP6 future model runs (e.g., changes in cloud cover, increased air temperatures, reduced sea ice thickness), visible light (PAR) reaching the surface water column will increase by 55-160% annually by the year 2100, based on climate scenarios SSP2-4.5 and SSP5-8.5 (Fig. 3, S3). Reductions in snow and sea ice thickness, and increased melt pond area due to warmer air temperatures, will also contribute to more light entering the water column (Fig. S1)."

Lines 95 – 96: Could you specify the statement about "increased melt ponds"? Do you mean "melt pond coverage"? And does the model output also show an increased melt pond coverage earlier in the year due to increased air temperatures?

The wording was not precise and have been changed. The melt pond model responds to the changes in air temperature by functionally reducing the albedo. The approach is the same as used in the CESM3 model. We quantified the impact of melt ponds on OSA, PAR, UV and other variables as part of our sensitivity tests.

Line 98: Will it increase the heat content of the "surface" water column? I'm not sure how much it affects deeper water layers?

We have not quantified how the surface heating will be mixed into the water column. Given the heat capacity of water we would expect heat to be distributed vertically due to turbulent mixing from wind and tides. We changed the wording to include "surface".

Line 99: "positive feedback loop"

Corrected

Lines 104 – 106: When talking about model output, I think observed changes should be still described a bit more cautiously as predictions. E.g Instead of describing results as "By 2050, there will be an estimated 25-30% increase in annual average PAR" it should be written as "By 2050, annual estimated PAR is estimated/ predicted to increase by 25-30%....". This applies for several other observations in this manuscript that are worded as definite future truths (e.g. Lines 149 – 150).

Thank you for the observations. We believe we have corrected for this now.

Line 113: "as well as it can restrict phytoplankton growth."

Section "UV-B under climate change": This section includes a lot of result interpretation that should be moved to the discussion to make it easier for the reader to focus on the results. Also, references to figures are usually stated at the end of the sentence.

We agree and have split this section into two where one part remains in Results and the other was integrated into the Discussion.

Line 137: Is there any "non-biological" chlorophyll a?

Good point. We removed the word "biological".

Lines 140 – 142: This sentence should be reworded. It's very hard to read. It should be also worded

as full names, Northern Bering/Chukchi Sea and Barents Sea, when % values are given for both. Currently it is stated as Northern Bering/Chukchi without "Sea"

Agreed and the sentence has been split into several shorter ones easier to read: "Our results reflect this variability. Under climate scenario SSP2-4.5, projected CMIP6 chlorophyll-a values suggest increases in annual average chlorophyll-a content of $2.5 \pm 14.5\%$ in the Northern Bering/Chukchi Sea and $11.7 \pm 16.7\%$ in the Barents Seas. We estimated these values as the annual averages from the period 1980-2000 relative to 2080-2100."

Lines 142 – 143: decrease/increase in which parameter?

We added the words "chlorophyll-a content" to indicate the parameter that decreases/increases. The sentence now says: "Conversely, there is a decrease of $-3.7 \pm 1.9\%$ in chlorophyll-a content for the Northern Bering/Chukchi Sea and an increase of $7.7 \pm 13.7\%$ in the Barents Sea for the same periods under SSP5-8.5".

It occurs to me that the shown historic negative trends in chlorophyll-a in Figure 6 for both regions do not match the observed historic positive trends in chlorophyll concentration from remote sensing observations (Lewis and Arrigo 2020). Does the biological model include under-ice blooms and lateral biomass advection from other regions that are causing the current observed overall increase in chlorophyll- a in the Arctic Ocean?

The observed trends by Lewis and Arrigo 2020 were based on satellite observed and calculations between 1998 and 2018 averaged annually. Our monthly differences between 1980-2000 and the following decades 2000-2010 and 2010-2020 indicate specific months where we observe a loss in chl-a, but also months where we see an increase. Overall, comparing each LME between 1980-2000 and into the future 2080-2100 we do observe an increase in chl-a for both LMEs under SSP245, but under SSP585 we see an increase in the Barents Sea and a smaller decrease in the Bering/Chukchi Sea. The uncertainty in these estimates is high and varies between climate models. Overall, the results do agree with Arrigo and Lewis 2020 for the Barents Sea but less for the Bering/Chukchi Sea when averaged annually for the historical decades. Still, there is high uncertainty in biological production estimates from climate models. We have added another sentence summarizing this in the results section: "Overall, we find there is high uncertainty in estimates of the future chlorophyll-a production." Our hope is that future climate models will include improved and more complex biological ocean modules that can provide improved projections with a higher degree of certainty.

Line 153: Omit the dot at the end of the sub-heading

Corrected

This section about impacts on fish also includes quite a bit of result discussion that could be moved to the discussion section so that it is easier to identify the model results of this study.

We agree and have moved parts of the text to the Discussion.

Line 184: Missing "under SSP5-8.5"

Corrected

Lines 184 – 187: If values in brackets are referring to the Northern Bering/ Chukchi Sea, then the months describing the seasons should not be given in brackets as well. Otherwise, it is very confusing.

Corrected

Line 195: " $\mu\text{mol photons m}^{-2} \text{s}^{-1}$ "

Corrected

Lines 201 – 216: It is very difficult to connect this paragraph with the previous paragraphs because it's suddenly a mix of statements about the PAR threshold and the temperature threshold. I think this paragraphs needs a bit more structure, so that it is possible to follow the story.

We agree with the reviewer. The section was edited substantially, and we hope it reads better now.

Line 221: It's better to say "positive feedback loops"

Corrected

Line 222: Maybe reword to "making it more difficult for long-lived marine species to adapt".

Corrected as suggested.

Line 223: "increased light levels"
Corrected as suggested.

Lines 225 – 227: Although this statement about a mismatch between energy requirements and light availability for visual feeding is further explained three paragraphs down, it seems to not be well connected in this section. I think the flow of this discussion section could benefit from restructuring to connect the statements in this paragraph and the next with the explanations in the paragraphs below. We agree and completely rewritten. We merged the first and third paragraph together to make these more coherent. We also removed some sentences that were repeated.

Lines 234 – 235: This sentence repeats the information of the previous sentence and is not needed. We agree, and removed the sentence.

Line 250: Changes to who's productivity? Phytoplankton, zooplankton, fish?
Corrected by adding "phytoplankton and zooplankton"

Lines 253 – 254: Earlier peaks in prey abundance due to earlier phytoplankton blooms? What impact do fall phytoplankton blooms, which are not uncommon in the Barents and Bering Sea (Ardyna et al. 2014), have on prey availability?
Good point. We added a sentence on this topic with reference to Ardyna et al. 2014 in the first paragraph of the Discussion.

Lines 256 – 258: How does the decrease in ice cover and increase in UV affect Polar cod egg survival?
Good point. We have calculated the daily exposure of eggs and compared the daily values with threshold reported in the literature and found values to be low (or moderate, but not high). We have added a sentence about this in the discussion. We did not include additional figures or results for uvb exposure as we feel the results were not that interesting to show and we have so much information already.

Lines 280 – 283: The statement about Polar cod seems to not fit well into this sentence. This could be a separate sentence that also provides percentages (or change in percentages) of polar cod abundance from these surveys for more context.
We rewrote sentence and removed "...if Polar cod disappeared from the.."

Line 285: The term "cold-water pool" should be explained more.
Agreed. We added two sentences that explain the cold-pool and reorganized the wording. We also added a new reference to Clement et al. 2022.

Line 291 "for a poleward full-year range expansion..."
Although changes in chlorophyll-a from the model output are presented in the result section, they are surprisingly not mentioned in the discussion. Could the authors discuss how the predicted changes might impact prey availability and connected to that the abundance, growth and survival of key fish species?
Thank you for the suggestion. We have included a few more sentences discussing the future changes in prey availability in the Arctic waters.

The presented predictions of changes in UV-B that reaches the ocean are also not mentioned in the discussion. However, as highlighted above, the impact of UV-B on fish species is discussed in the results section, which I think should be moved into this section to provide a more in-depth discussion on how changes in ocean light may affect fish.
We agree and have added a wider discussion and references on the impacts of UV-B on marine ecosystems.

Lines 308 – 311: Could you break this very long and cumbersome sentence into two sentence, which will make it easier to read. Also, what were the 11 required variables mentioned here?
We have rewritten this section and improved the sentence. The variables required (12) are listed further down in the Methods section.

Line 309: Could you define "RTM" here again?

Fixed.

Line 320: "from one to another"

Removed.

Line 330: "and in the model independence..."

Fixed.

Line 356: Where was the monthly average cloud data downloaded from or was it simulated? Please provide more information.

The monthly averaged cloud data are available as a standard CMIP6 output and was downloaded from the Google Cloud CMIP6 repository same as the other variables. We added a sentence stating this data is available from CMIP6.

Line 357: Does the correction factor reference to the rho fraction?

Yes. We corrected the sentence.

Lines 406 – 408: The sentence currently reads "Not all climate models [...] do not provide the total column ozone...". Is this the correct wording?

Yes almost we added the word "therefore" (...and therefore do not provide...)

Line 438: "melt ponds"

Fixed.

Line 443: The value for fh is not provided.

Fixed

Line 443: "To represent the impact of melt ponds on albedo,...."

Fixed

Lines 443 – 447: How was the change in melt pond coverage calculated in the models?

We have added more details on how melt ponds were calculated.

Line 456: I think that the assumption of pure ice could introduce a considerable error in calculated under-ice light availability in spring when the ice algae layer can strongly absorb light within the ice layer, especially in the blue and red spectrum (Ehn and Mundy 2013, Perovich 1996). Given the patchiness and variability in ice algal biomass, the changes in snow depth probably have a much larger impact on changes in light transmission on a pan-Arctic Scale. However, I think the reasoning for excluding a chlorophyll layer in the ice cover in spring should be briefly mentioned here.

We agree and have added a sentence "We also do not account for attenuation caused by sea ice algae, as the CMIP6 models do not provide information." We have referenced Ehn and Mundy 2013.

Line 486: "and UV was integrated from 280-400nm."

This sentence was changed.

Lines 496 – 500: I disagree that changes in pigment concentration will only have a small impact on light availability in the water column. I agree that subsurface maxima are a common feature in the Arctic summer months. But this manuscript highlights that light availability will particularly increase in the spring and fall months due to the thinning and disappearing sea ice cover. During these periods, large under-ice blooms, especially in the Chukchi Sea, and fall blooms, a common feature in the Barents Sea, can have a large impact on light availability and visibility for visual hunters.

We agree and have added a sentence to highlight this missing feature of the model: "Still, under-ice blooms during spring could impact the attenuation of light reaching the water column, which the model does currently not account for.". The open ocean fall bloom production in the Barents Sea is accounted for using the chlorophyll-a values from CMIP6.

Figures

Figure 1: Currently it looks like as much light is transmitted through bare ice as it is through thick

snow-covered ice. That's not really true (the transmitted arrows are equally thick). I also think it would be more accurate to call the green cells "phytoplankton" instead of just "chlorophyll".

This figure is meant as a simple illustration. We have kept the figure as it was because it's not so easy to change, but if the manuscript is accepted, we will do the modifications you suggested.

Figure 3: Please define PAR in the figure caption again (photosynthetically active radiation). Also, the results section does not explain the sudden significant increase in PAR after 2050 for the SSP5-8.5 scenario in the Northern Bering and Chukchi Sea. Why is there such a large increase? When combining this plot with the content of the supplementary figures, the year 2050 seems to generally be a tipping point in SSP5-8.5. Can this be explained more in the manuscript?

We agree and also find this interesting. We have added multiple sentences regarding these observations to the Results section and also added some discussion around these observations. We appreciate the comment.

Figure 5: Impacts of temperature, ice concentration and light on egg survival and larval growth potential; not just temperature and light according to the y-axes. The figure caption has some grammatical issues in the description of what is shown on each y-axis. Additionally, the y-axes of the middle column of sub-plots are very cramped in.

We have fixed the caption. The figure took a long time to get correct and we have not modified it further.

Figure 6: I recommend to label the x-axes "Change in Temperature" and "Change in Chlorophyll"
Fixed.

Line 758: "are shown"

Fixed.

Figure 7: The axis labels are very small, hard to read. And the figure caption does not describe what the solid and dotted lines are showing.

Fixed by increasing font size for axis.

Line 765: "monthly average PAR"

Fixed.

Supplemental figures:

Figure S1: Chlorophyll subplot – The y-axis unit "1e-7" is much too small. Also please label it Chlorophyll-a

Fixed.

Figure S2: The term "predictable water" is uncommon. What does it mean?

We have added a definition in the caption and in the main manuscript. It is a term used in the CMIP6 world meaning "the total amount of water vapor present in a vertical column of the atmosphere"

Figure S3: Axis labels are too small. Also, the resolution of this plot is very low. When zooming in to better read the lines and labels, the plot becomes very blurry.

Figure S4: The term "PAR" should be used instead of "Ocean light" to be more clear about the wavelength range and to match the plot label.

Fixed.

Figure S5: Some verbs are missing in the sentences of the figure caption.

Fixed.

Figure S6: The axis labels are too small again. And the citation lists "upper panels", but there are no other panels in this plot.

Fixed.

Figure S7: How is this figure different from Figure 6 in the manuscript?

Figure 5 (previously 6) in the manuscript shows values for SSP2-4.5 and figure S6 (previously S7) shows for SSP5-8.5.

Figure S8: "Functions are shown for a)..."

Fixed.

Reviewer #1 (Remarks to the Author):

I thank the authors for spending time reviewing their manuscript based on the comments from all reviewers.

We appreciate the feedback, and we are grateful that the effort revising the manuscript was well received.

A much needed sensitivity analysis has been added as well as comparison of the model outputs to reanalysis. This helps the reader assess the model uncertainty.

We fully agree that the sensitivity analysis provided needed insight into the RTM and its components and overall improved the outputs of this study.

I believe that all comments from reviewer 1 and 3 have been addressed. However, I still express some doubts as to how the authors justify some of their decisions (i.e., limiting runs because files become too large...).

Regarding the remarks from reviewer 2, I acknowledge that the authors have tried to answer concerns or to provide answers when they disagreed with the comments. The authors justified their use of a limited number of CMIP6 models which is fine. I struggle to understand their choice of ensemble members across models though. I believe that the authors justified themselves by saying that "By utilizing multiple ensemble members across various climate scenarios, we aim to capture a broad range of potential outcomes." which is true but by selecting some ensemble members over others without truly justifying it, they introduce a form of bias to their results.

The ensemble members were chosen due to the availability of the 12 variables across individual members required to run the RTM, and their frequent use in climate literature (Long *et al.*, 2021; Liu *et al.*, 2022; Meucci *et al.*, 2024). This is a standard practice in comparable studies using CMIP results. We do not see how using a subset of all produced ensemble results for each ESM could introduce a significant statistical bias given that: 1) we are aggregating results over multiple ESMs and ensemble members, 2) the differences between ensemble members are mostly driven by random factors with no harmonization between ESMs, 3) there is no reason (to our knowledge) to suspect that the above data availability and frequent use criteria would favor ensemble members with certain behaviors.

Most of my worries however, reside in the fact that aside from using different realisations between models, the authors also used different forcing and physics settings (see Table S1) which are not consistent between models. How can results therefore be analysed together or compared? Are the authors aware of this?

We believe that the concern over the use of different forcing and physics applies to two ESMs in Table S1: for CanESM we use both 'p1' and 'p2', and for UKESM we use only 'f2' rather than the standard 'f1'. For all other datasets we use the standard configuration 'rXi1p1f1' where only random factors are varied.

In the case of CanESM5 (CanESM5.0.3), the two physical configurations differ only in the remapping method used for the wind fields (bilinear vs. conservative, see (Swart *et al.*, 2019),

Table 2) which results in minor differences, strongest over Antarctica (Swart *et al.*, 2019, Appendix E). The ‘p2’ physics basically removed a bug in ‘p1’ that led to cold spots over Antarctica (Swart *et al.*, 2019; Sigmond *et al.*, 2023). Hence, the use of ‘p2’ for CanESM5.0.3 fixed problems in ‘p1’ and therefore these ensemble members are more widely used. Still, by including ensemble members with both ‘p1’ and ‘p2’ physics we explore slightly more of the underlying modelling uncertainty. Since our results are aggregated over multiple ESMs we do not see how this could bias our results.

In the case of UKESM1-0-LL, the ‘f2’ configuration is in fact the standard configuration because the ‘f1’ setting is actually not used (see (Yool *et al.*, 2021), or <https://ukesm.ac.uk/cmip6/variant-id/>).

This information has been summarized and added to the Methods section of the manuscript under “**CMIP6 model selection**”:

“Regarding the choice of ensemble members (Table S1), for the CanESM5 model we use both ‘p1’ and ‘p2’, and for UKESM we use ‘f2’ rather than the standard ‘f1’. For all other datasets, we use the standard configuration ‘rXi1p1f1’ where only random factors are varied. For CanESM5, the two physical configurations differ in the remapping method for wind fields (bilinear vs. conservative, see (Swart *et al.*, 2019), Table 2), resulting in minor differences, strongest over Antarctica ((Swart *et al.*, 2019), Appendix E). Including ensemble members with both ‘p1’ and ‘p2’ physics allow us to explore more of the underlying modelling uncertainty. For the UKESM1-0-LL model, the ‘f2’ configuration is the standard as the ‘f1’ setting is not used ((Yool *et al.*, 2021)).”

A minor detail but I would still suggest that the authors carefully proof read their manuscript as I noticed a few typos here and there.

We have carefully read through the manuscript and tried to eliminate all typos.

We hope that the added information and the edits are satisfactory.

References:

Liu, H. *et al.* (2022) “An ocean perspective on CMIP6 climate model evaluations,” *Deep-sea research. Part II, Topical studies in oceanography*, 201, p. 105120.

Long, M. *et al.* (2021) “Multi-aspect assessment of CMIP6 models for Arctic sea ice simulation,” *Journal of climate*, 34(4), pp. 1515–1529.

Meucci, A. *et al.* (2024) “An 8-model ensemble of CMIP6-derived ocean surface wave climate,” *Scientific data*, 11(1), p. 100.

Sigmond, M. *et al.* (2023) “Improvements in the Canadian Earth System Model (CanESM) through systematic model analysis: CanESM5.0 and CanESM5.1,” *Geoscientific model development*, 16(22), pp. 6553–6591.

Swart, N.C. *et al.* (2019) “The Canadian Earth System Model version 5 (CanESM5.0.3),” *Geoscientific Model Development*, 12(11), pp. 4823–4873.

Yool, A. *et al.* (2021) “Evaluating the physical and biogeochemical state of the global ocean component of UKESM1 in CMIP6 historical simulations,” *Geoscientific model development*, 14(6), pp. 3437–3472.

Review of manuscript entitled “Climate change impacts on ocean light in Arctic ecosystems” by Kristiansen et al.

Having reviewed a different version of this manuscript before, I thank the authors for carefully considering previous remarks in this new version they submitted. The manuscript is still focused on the use of a newly developed method to estimate the propagation of solar radiation from the top of the atmosphere to the bottom of sea ice, how the light availability and warming might change in the future and the impact on cold-water fish species.

I appreciate the work done by the authors to expand their studies to include more CMIP6 models and realisations to offer the reader a broader understanding of the inherent variability between each model and their variants. The study is, I think, interesting but some details are still lacking in my opinion to have it ready to be published.

I would say that the one of the primary issues resides in a proper parameterisation of the light attenuation by snow and sea ice despite the emphasis of the study on future changes in snow and sea ice condition (see comment for this paragraph). Secondly, as this is a modelling study, I think it would benefit from more comparison to in situ datasets regarding actual levels of light and parameterisation, this is very much lacking now.

I'm also still a bit puzzled by the RTM. The authors refer to it as a “newly developed RTM” and now properly explain the different parts in the Method section, but it appears that this RTM is simply different published algorithms assembled. The concept of a RTM from toa to bottom of sea ice is interesting and I acknowledge that there is considerable work behind this study, but, in my opinion, with the focus of this manuscript on change in snow and sea ice conditions, a stronger emphasis should have been put on the modelling of light transmission through snow and sea ice and this part is underwhelming.

Please see my various comments below:

Line 30-31: I'm not sure I agree with this sentence. More and more studies have been published over the years that are focused on the response of ecosystems to change in light availability in the Arctic, sure it's still a challenging topic but lots of progress has been done, see Tedesco et al., 2019; Connan-McGinty et al., 2022 or the recent Flores et al., 2023 which it seems you might have not seen.

Line 41-45. Is it newly developed because it combines various algorithms that were not associated before ?

Line 160: correct SSSP2-4.5

Line 180: I think it would be very interesting to compare your results to Flores et al 2023. They did a similar study for the whole Arctic looking at light levels for zooplankton vertical migration from an observation point of view but also by using CMIP6 future projections.

Line 195. Interesting, first time I see you now use the proper unit for PAR when used for biological purposes. Although I do understand that you want to reach a broader audience by using the physical unit $W m^{-2}$ for PAR, the work in this manuscript will be of greater interest for scientists interested in biological outcomes of increase in light availability and they are usually more used to the proper unit (i.e. you had to use it in parenthesis at some point to compare to other studies).

Line 307.

Regarding models, I would like to bring your attention to the sensitivity study from Dirk Notz on Arctic sea ice within CMIP6 (Notz 2020) where a certain number of CMIP6 models were found to properly simulate sea ice loss while also properly representing global temperature increase over time. Two of the models you used are in the list but not the other twos.

Line 346. I understand that you used several CMIP6 variables such as cloud cover and ozone layer combined with the amount of solar radiation reaching the top of the atmosphere to simulate the proportion of sunlight reaching the snow/sea ice/ ocean surface. Did you try to consider the downwelling shortwave at the surface variable from these same models and compare with your method ? and if not, why not ? Please explain.

Line 380-382. I find it awkward to use monthly (and not daily) outputs but then calculate every 4 hours for the middle of the month. Light is such a sensitive variable to daily processes. What is the reasoning behind all this ? Is it because it might be computationally too heavy ? Please explain.

Line 385-391 This whole paragraph is a repetition of a previous paragraph. There is quite some repeating in the method section, you might want to avoid that.

Line 435-438. There is no problem in using pure snow or sea ice properties, however this sentence (here and elsewhere) put the emphasis on the impact of soot particles which you are not considering. If you desire to keep it like that, you should add a few lines in the discussion about the impact of not considering soot particles.

Line 438-439. Why not look for measured albedo values instead of modelled ones ? There are so many *in situ* studies on albedo of Arctic sea ice. You'd then see that an albedo of 0.96 for snow covered sea ice happens with (very) fresh snow and it is rarely seen *in situ*. It is even less seen in a melting Arctic (i.e., Perovich and Polashenski 2012). Albedo reduces because of melt pond of course and it is good that you account for that, but it also reduces because of snow metamorphism over time. Please explain.

Line 443. Missing value: $F_h = ?$

Line 453-458. I'm kind of surprised regarding the attenuation through snow and sea ice. Several studies covering *in situ* and remote sensing data as well as model based have been using the exponential decay relationship to evaluate solar partitioning by snow and sea ice before, this is not new, and it seems what you've used is even less complex than these studies (i.e., Light et al., 2008, 2015, Lebrun et al., 2023, Stroeve et al., 2021) as not considering a drained layer in sea ice or even the

existence of a surface scattering layer forming due to interaction of solar irradiance with the top layer of sea ice. It is a bit underwhelming considering that the study is focused on the change in snow and sea ice conditions on under sea ice light availability.

Regarding the extinction coefficient in sea ice: it is mentioned that it's a spectral extinction coefficient but no mention of the values nor where it is coming from ?

Regarding the extinction coefficient in snow: I am very surprised of the value used of 0.2 m^{-1} . It is extremely small, even way lower than the usual broadband extinction coefficient for sea ice commonly used. It is even more curious considering the albedo of 0.96 that you used. Where is this value originating from, is it for a single wavelength ? It is a broadband value as well which makes it even more confusing that it's so low. Please explain the reasoning and references behind this.

I would suggest you have a look at the literature for extinction coefficients in snow (i.e Perovich 1996 or the recent paper by Lebrun et al., 2023). I would also suggest considering different conditions for snow evolution stages where albedo and extinction coefficient change accordingly.